# Functionally analogous body- and animacy-responsive areas are present in the dog (*Canis familiaris*) and human occipito-temporal lobe

Magdalena Boch [1,2✉], Isabella C. Wagner [1,3,4], Sabrina Karl[5], Ludwig Huber[5,6] & Claus Lamm [1,3,6]

Comparing the neural correlates of socio-cognitive skills across species provides insights into the evolution of the social brain and has revealed face- and body-sensitive regions in the primate temporal lobe. Although from a different lineage, dogs share convergent visuo-cognitive skills with humans and a temporal lobe which evolved independently in carnivorans. We investigated the neural correlates of face and body perception in dogs ($N = 15$) and humans ($N = 40$) using functional MRI. Combining univariate and multivariate analysis approaches, we found functionally analogous occipito-temporal regions involved in the perception of animate entities and bodies in both species and face-sensitive regions in humans. Though unpredicted, we also observed neural representations of faces compared to inanimate objects, and dog compared to human bodies in dog olfactory regions. These findings shed light on the evolutionary foundations of human and dog social cognition and the predominant role of the temporal lobe.

[1] Social, Cognitive and Affective Neuroscience Unit, Department of Cognition, Emotion, and Methods in Psychology, Faculty of Psychology, University of Vienna, Vienna, Austria. [2] Department of Cognitive Biology, Faculty of Life Sciences, University of Vienna, Vienna, Austria. [3] Vienna Cognitive Science Hub, University of Vienna, Vienna, Austria. [4] Centre for Microbiology and Environmental Systems Science, University of Vienna, Vienna, Austria. [5] Comparative Cognition, Messerli Research Institute, University of Veterinary Medicine Vienna, Medical University of Vienna and University of Vienna, Vienna, Austria. [6]These authors contributed equally: Ludwig Huber, Claus Lamm. ✉email: magdalena.boch@univie.ac.at

The ability to perceive others is essential for successful social interactions and survival. Faces and bodies of others convey a wealth of social information, enabling an observer to infer, for instance, others' emotional states or their intentions[1–3]. Consequently, accurate perception of faces and bodies is a particularly important building block for social perception.

Decades of neuroimaging research in humans revealed a predominant role of the occipito-temporal cortex for perceiving others with distinct, but adjacent regions specialized for face and body perception as part of the ventral visual pathway[4]. The human ventrolateral visual cortex responds to animate compared to inanimate stimuli more generally[5] and encompasses face- and body-sensitive regions in the lateral occipital cortex (occipital face and extrastriate body area) and inferior temporal cortex (fusiform face and body areas[6,7]; or see ref. [8] for review). This category-preference can also be observed in the extrastriate body and fusiform face area of human infants[9], and distinct neural representations (i.e., increased similarity of multivariate activation patterns) for faces and bodies were shown in the ventral visual pathway of human infants even in the absence of category-specific brain regions[10].

Comparative research in non-human primates further revealed homologous regions for face and body perception[11–14], emphasizing the role of the primate temporal lobe in processing and integrating social information (see e.g., ref. [15], for review). Regarding the perception of another individual's species-identity, preference for conspecific faces seems to be less pronounced in humans than in non-human primates[12,16]. However, research outside the primate lineage on social perception in general and on face and body perception is scarce. This limits our knowledge on the evolution of these phenomena and their neural underpinnings.

Dogs (Canis familiaris) are a particularly promising and emerging novel model species in this respect. Although from a completely different lineage, they share numerous analogous visuo-cognitive skills with humans and other non-human primates. For example, dogs can differentiate between faces[17] and discriminate facial emotional cues of humans and conspecifics[18,19]; they even over-imitate their caregiver's (irrelevant) actions[20,21] and demonstrate further complex behaviors, such as visual perspective taking[22,23]. Dogs have been humans' closest companion since thousands of years[24] and their evolution has been shaped by humans through domestication[25,26]. They have been exposed to the same (visual) environment, allowing for comparative task designs with humans using the identical stimulus set. Dogs are also highly receptive to training[27,28], which allows for non-invasive neuroimaging studies with fully awake and unrestrained pet dogs. Finally, dogs have a temporal lobe, which evolved in carnivorans independently to primates[29,30]. Thus, comparative neuroimaging studies with dogs and humans present an excellent opportunity to test the evolutionary history of the human and dog social brain and the role of the temporal lobe[31,32].

Research so far suggests an involvement of dogs' temporal lobe in face perception, but inconclusive results have triggered a debate on whether the occipito-temporal specialization for face perception in dogs matches that of humans[33–37]. Apart from one electroencephalography (EEG) study[38], prior neuroimaging studies did not find greater activation for faces compared to scrambled images[33,34], but compared to scenes[34] or objects[34,36,39], or didn't have any non-facial controls[35], questioning if face-sensitivity rather reflects differences in low-level visual properties. Further, almost all prior studies lacked animate stimuli other than faces[33–36,38,39] and the only study[37] with another animate stimulus category (i.e., the back of the head) had no inanimate control condition. Thus, studies so far could not control for animacy as an alternate explanation of the supposed face-sensitive responses. Results regarding species preferences were also mixed, ranging from no conspecific-preference[34] to separate regions for dog and human

face perception[35] and a recent report of a conspecific-preferring visual region[37]. Therefore, previous studies carry several limitations that prevent a better understanding of how dogs perceive others compared to humans: they cannot disentangle face-sensitive from general animate vs. inanimate perception, are inconclusive regarding perception of con- and hetero-specific individuals, and provide limited insights into potentially convergent neural underpinnings of face perception, with only two comparative studies so far[37,39].

The fact that no previous work investigated the neural bases of body perception in dogs is another major research gap. Bodies play an important role for social perception in general but especially for dogs, who show high responsiveness to gestural or referential bodily cues[40–44]. In this context, it is particularly noteworthy that dogs even outperform humans' closest living relatives, chimpanzees (Pan troglodytes), in utilizing human gestural cues[45]. Such behavioral evidence and considering the link between extensive visual expertise and engagement of dedicated higher-order visual areas in humans (see ref. [46] for recent meta-analysis) allows the testing of unique hypotheses, such as that dogs would show involvement of higher-order visual regions in body perception.

The aim of the present comparative neuroimaging study was to investigate the neural bases of animate vs. inanimate and face vs. body perception, and their potential convergent evolution in the dog and human brain. For our main analysis, we employed a functional region-of-interest (fROI) analysis on the individual level to investigate face- and body-sensitivity in the occipito-temporal cortex of dogs and humans and whether these regions responded differently to conspecific vs. heterospecific stimuli, as indicated by differences in activation levels. This was complemented by whole-brain analyses of multivariate activation patterns (representational similarity analysis) to identify regions that might not be category-sensitive in terms of higher activation levels but show increased neural pattern similarities for faces, bodies, inanimate objects, or conspecifics and heterospecifics.

As predicted, we found a body-sensitive area in the dog occipito-temporal lobe, emphasizing the link between visual expertise and engagement of higher-order visual areas. The analysis also revealed analogous occipito-temporal brain areas sensitive for animate entities (i.e., faces or bodies) compared to inanimate objects and low-level visual controls in dogs and humans indicating a convergent evolution of these neural bases. However, we only detected face-sensitive areas in humans. This suggests that previously identified face-responsive areas in the dog brain may respond more generally to animate compared to inanimate stimuli. Contrary to what we observed in humans, dog body- and animate-sensitive brain areas did not show a preference for conspecifics compared to humans which complements the large body of behavioral evidence for dogs' high-sensitivity towards both dog and human whole-body and facial visual cues[19,47,48]. While unpredicted, we also found increased pattern similarities for conspecific compared to human bodies and faces compared to inanimate objects in dog olfactory regions. Considering the direct pathway connecting the dog visual and olfactory cortices[49] this likely reflects dogs' high olfactory sensitivity and its interplay with visual perception to navigate their social environment (see ref. [50] for review). Overall, the results underline the predominant role of the temporal lobe for social information processing in two distant mammalian species, alongside species-specific adaptations potentially reflecting the ecological significance of visual facial information for humans and olfaction for dogs.

## Results

Fifteen awake and unrestrained pet dogs (Fig. 1a) and forty human participants underwent functional magnetic resonance imaging (fMRI) while viewing images of human and dog bodies, faces, and

**a**   Set-up dog participants

**b**   Image categories

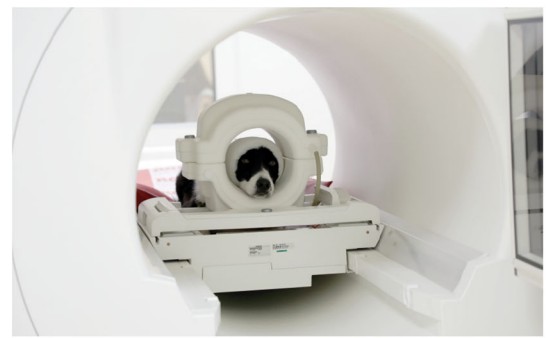

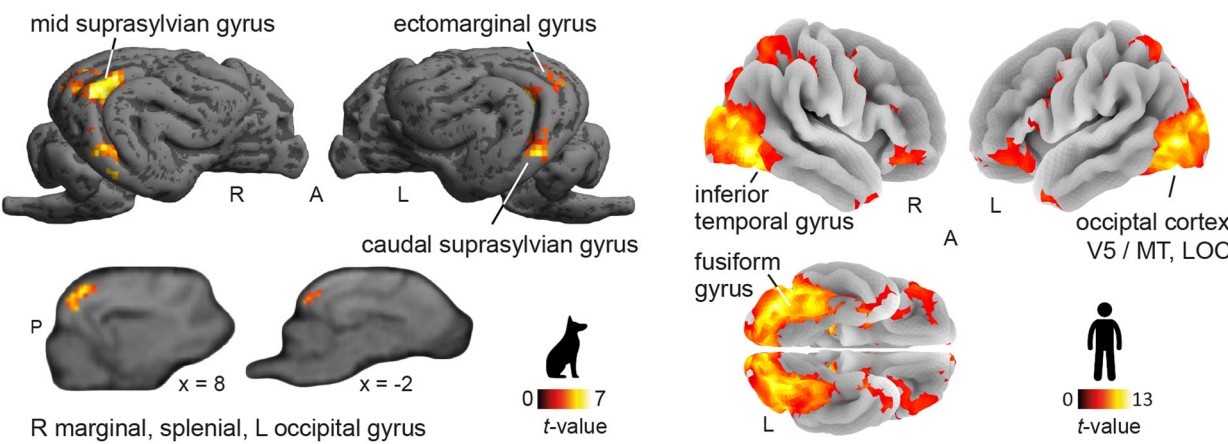

**c**   Visual-responsive brain areas (localizer data set: all stimuli > baseline)

**Fig. 1 Overview on comparative experimental approach and visual-responsive areas in occipito-temporal cortices of both species. a** We obtained all imaging data using a 3 T Siemens Skyra MR-system, equipped with a 15-channel human knee coil to scan the dogs and a standard 32-channel head coil for the human participants (not depicted). Dogs had received extensive training[28] to stay motionless without restraints or sedation and could leave the MR scanner at any time via a custom-made ramp positioned at the scanner bed. They had an additional head bandage to secure optimal positioning of the earplugs throughout the scan session. **b** Fifteen dogs and forty humans viewed images of human and dog bodies, human and dog faces, inanimate objects, or grid-scrambled versions of these images presented in a block design (see section Stimulus material for details). **c** Using data from the first task run (i.e., localizer data set) we identified visual-responsive areas (i.e., all stimuli > baseline) in the occipito-temporal cortices of both species (see Supplementary Table S1 for detailed areas). The gyri containing visual-responsive activation served as anatomical search spaces for the functional regions-of-interest analysis of the dogs, for humans we used parcels derived from a previous study[52] (see Fig. 2a). Results are $p < 0.05$ FWE-corrected at cluster-level using a cluster-defining threshold of $p < 0.001/0.005$ for humans/dogs. Anatomical nomenclature for all figures refer to a dog anatomical atlas[53] normalized to a breed-averaged template space[83], the human brain atlas from the Laboratory for Neuro Imaging (LONI) Brain Atlas[96] (LPBA40, http://www.loni.usc.edu/atlases/) and the Automatic Anatomical Labeling atlas[97] (AAL2). R right, L left, A anterior, P posterior, MT middle temporal visual area (V5), LOC lateral occipital cortex. Example category images in (**b**) are license-free stock photos derived from www.pexels.com and were modified for the study purpose (i.e., head or body cut out); the dog and human icons in **c** were purchased from thenounproject.com (royalty-free license).

inanimate objects (e.g., different toys), with grid-scrambled versions of these images serving as controls for low-level visual features (Fig. 1b). Over the course of two 5-min runs, participants saw 180 different images presented in a block design and blocks were separated by an implicit visual baseline.

We analyzed the dog fMRI data with a tailored haemodynamic response function (HRF) shown to significantly improve neural signal detection power[51]. All statistical tests were corrected for multiple comparisons.

**Visual-responsive voxels and category-sensitive functional regions-of-interest to test category-sensitivity.** First, we investigated whether dogs and humans have comparable and specialized cortical regions for face and body perception. We employed a

functional region-of-interest (fROI) analysis approach (see e.g., ref. [9], for recent application in human infant fMRI) which relied on splitting the data into two independent data sets: (a) a localizer data set (first task run) to define individual potential face- or body-sensitive areas in visual-responsive brain regions (Figs. 1c, 2a) and (b) a test data set (second task run) to extract activation levels from these regions. We chose this approach for two main advantages. First, defining individual fROIs accounted for differences in the location of activation peaks between participants (as reported in previous studies[35,37]). Second, this allowed us to not only localize potential face- or body-sensitive regions but also to directly evaluate their category-sensitivity using the left-out data.

For each participant, we defined bilateral fROIs within constrained search spaces to preserve spatial information (i.e., the rough anatomical location of activation peaks). For the human

participants, we used previously reported anatomical regions in the ventral visual pathway known to be engaged in face and body perception[52] as search spaces: the extrastriate body area, fusiform body area, occipital face area, and fusiform face area (Fig. 2a). For the dog participants, we could not build on previous work, as body perception has not been studied yet in previously published work, besides other reasons (lack of a universally shared template space such as the human MNI space in dog neuroimaging, use of unpublished templates, no reports of peak coordinates, and/or unavailability of data). Therefore, we first compared activation levels associated with the visual presentation of all stimuli compared to implicit baseline (i.e., white cross presented on gray background) using the localizer data set. This revealed visual-responsive activation in the occipital-, splenial-, ectomarginal-, caudal-, mid suprasylvian- and marginal gyri (Fig. 1c, Supplementary Table S1). We then used anatomical masks[53] of these gyri to serve as search spaces for the dog participants (Fig. 2a). For comparison between species, we also report visual-responsive brain areas for the human participants (i.e., all stimuli > baseline), confirming involvement of the occipito-temporal cortex, including the lateral occipital cortex and fusiform gyrus (Fig. 1c, Supplementary Table S1).

We then restricted each search space to voxels responding stronger to faces compared to bodies and vice versa (i.e., face or body search spaces) and selected the top-10% voxels with the strongest activation levels for bodies or faces compared to inanimate objects to define the individual fROIs within the face and body search spaces (see Supplementary Table S2 for mean fROI sizes). Activation levels for conspecific and heterospecific stimuli were pooled (i.e., assigned equal weights to create beta maps) because the main focus of the study was to localize universal face and body regions.

Choosing the top 10% voxels was an a priori analytical decision we made based on the size of the resulting individual fROIs before any activation levels were extracted (see Methods: Functional region-of-interest approach for details). However, after we conducted the main analysis, we also extracted parameter estimates for a range of different percentage cut-offs between 1% to 100% to validate the results using this threshold, altogether confirming that the 10% threshold was an appropriate fROI size for detecting relevant activation levels (see Supplementary Note 1 and Supplementary Fig. S1).

**Overall sensitivity for animate stimuli, one body-sensitive region in dogs, and face-sensitive regions exclusive to humans.** Next, we extracted the mean activation signal during viewing of faces, bodies, and inanimate objects (all compared to scrambled controls to account for low-level visual features) from the individual fROIs using the test data set (i.e., beta maps) and performed group comparisons by running repeated measures ANOVAs. In dogs, we localized a body-sensitive region in the mid suprasylvian gyrus (i.e., greater activation for bodies compared to faces and inanimate objects; body fROI). We further observed significantly greater activation levels for faces and bodies compared to inanimate objects in four fROIs located in the extrastriate occipital cortex and in the temporal association cortex, but no differences between image categories in early visual cortex areas, such as the marginal or occipital gyrus (Supplementary Table S3, Fig. 2b). More specifically, the mid suprasylvian and ectomarginal face fROIs and the caudal suprasylvian body fROI showed greater sensitivity for animate compared to inanimate stimulus conditions, but no difference within the two animate categories (i.e., faces vs. bodies). In humans, we found evidence for both body- and face-sensitivity, with the strongest overall activation for bodies in the extrastriate and fusiform body areas and for faces (i.e., face-sensitivity) in the occipital

and fusiform face areas (Supplementary Table S3, Fig. 2b). In summary, the first analysis step relying on fROIs revealed multiple occipito-temporal regions that were responsive to animate stimuli in both species, first evidence for a body-sensitive region in dogs, and further sub-divisions into multiple face- and body-sensitive regions only in humans.

Since anatomical search space definitions and sample sizes differed between the two species, we conducted a control analysis with the human data using anatomical masks instead of parcels (thus matching the analysis in the dogs) and performed the analysis in 1000 randomly drawn sub-samples of $n = 15$ participants (i.e., the dog sample size). The results corroborated that the observed differences between dogs and humans (i.e., face-sensitivity) are also present when accounting for these methodological differences (see Supplementary Note 2 and Supplementary Figs. S2–S3 for detailed approach and results). A further control analysis also confirmed that the observed face-, body- or animacy-sensitivity in the dog and human brain cannot be explained based on differences in low-level visual properties (i.e., hue and saturation) between stimulus categories (see Supplementary Note 3 and Supplementary Fig. S4).

**Complementary whole-brain univariate analysis confirms body-sensitivity in both species but face-sensitivity exclusive to humans.** Next, we conducted an exploratory whole-brain analysis to complement the fROI analysis and investigate if we can detect the category-sensitive areas using whole-brain group comparisons. In dogs, the whole-brain one-way repeated measures analysis of variance (ANOVA; levels: faces, bodies, inanimate objects; all conditions >scrambled control) revealed greater activation for bodies compared to faces and inanimate objects (i.e., body-sensitivity) in the mid and caudal suprasylvian gyrus as well as an animacy-sensitive region (i.e., greater activation for faces and bodies > inanimate objects) in the mid suprasylvian gyrus (see Fig. 3a, Supplementary Table S4). We did not find face-sensitive regions (i.e., greater activation for faces > bodies, inanimate objects), even after lowering to a liberal cluster threshold of $k = 10$ voxels. In humans, we found face-sensitive regions in the lateral occipital cortex and fusiform gyrus and activation of the hippocampus during face perception. For bodies, we observed greater activation compared to all other conditions in the lateral occipital cortex, the fusiform gyrus as well as the superior and inferior parietal lobules, the middle frontal and precentral gyrus and the thalamus. The occipito-temporal and fronto-parietal regions were also active when we compared faces and bodies (i.e., animate stimuli) to inanimate objects (see Fig. 3b, Supplementary Table S5).

**Conspecific-preference in human extrastriate body and fusiform face area.** Next, we investigated whether sensitivity for faces or bodies differed when participants saw images of conspecifics compared to heterospecifics. To this end, we performed repeated measurements ANOVAs with animate stimulus (bodies, faces) and species (dogs, humans) as factors to test for potential main effects of species or interactions between species and animate stimulus. For the dogs, we focused the analysis on the four fROIs that showed a preference for bodies or animate stimuli conditions (see above) and did not find higher activation in response to conspecifics compared to humans (Supplementary Table S6). In humans, the analysis revealed a significant interaction between the animate stimulus condition and species, reflecting greater activation for human compared to dog bodies in the extrastriate body area, as well as a significant main effect of species in the fusiform face area with greater activation for human compared to dog images regardless of animate stimulus condition (i.e., faces and bodies; see Fig. 4 for post-hoc comparisons, Supplementary Table S6 for all ANOVA results).

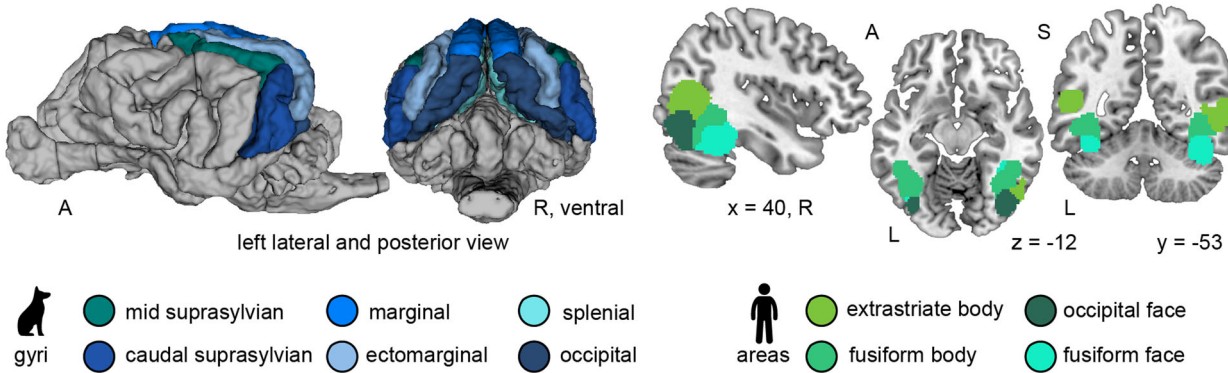

**a** Bilateral anatomical search spaces to create individual functional regions-of-interest (fROIs)

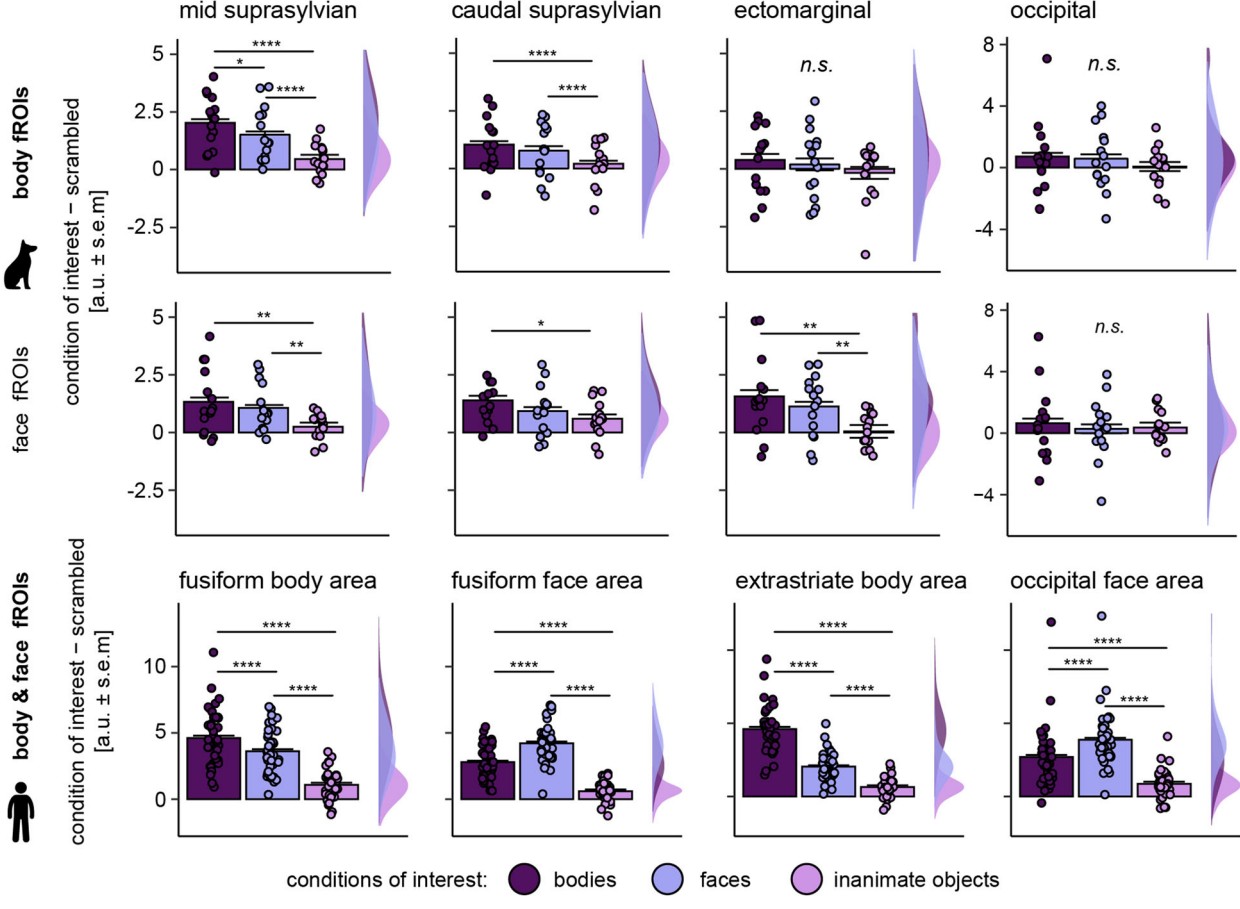

**b** fROI analysis: mean activation extracted from independent test data set

conditions of interest: bodies | faces | inanimate objects

**Fig. 2 Overall sensitivity for animate stimuli, one body-sensitive region in dogs, and multiple face- and body-sensitive regions exclusively in humans. a** Based on the localizer data set, we defined individual category-specific regions-of-interest (functional region-of-interest approach, fROI) within multiple anatomical search spaces using the contrasts bodies >faces and bodies >inanimate objects (body fROIs); and faces >bodies and faces >inanimate objects (face fROIs). **b** From these individual fROIs, we then extracted activation estimates during the viewing of bodies (purple), faces (lilac), and inanimate objects (pink; all compared to scrambled controls) using the data from the left-out data set. In dogs (N = 15), we observed greater activation levels for faces and bodies compared to inanimate objects in the extrastriate cortex (ectomarginal gyrus) and the temporal multimodal association cortex (mid and caudal suprasylvian gyrus; Supplementary Table S3). In addition, higher activation for bodies than faces in the mid suprasylvian body fROI provides first evidence for a body-sensitive region in the dog temporal lobe. Humans (N = 40), similar to dogs, showed strongest activation levels for animate stimuli in all fROIs. In contrast to dogs, human face and body fROIs showed sensitivity for the respective stimulus category (bodies > faces in body fROIs, faces > bodies in face fROIs). Planned comparisons were false discovery rate (FDR) corrected to control for multiple comparisons. *p < .05, **p < 0.01, ***p < 0.001, ****p < 0.0001, error bars represent the standard error of the mean (s.e.m.). a.u. arbitrary units. A anterior, P posterior, S superior, L left, R right. The dog and human icons in panels a-b were purchased from thenounproject.com (royalty-free license).

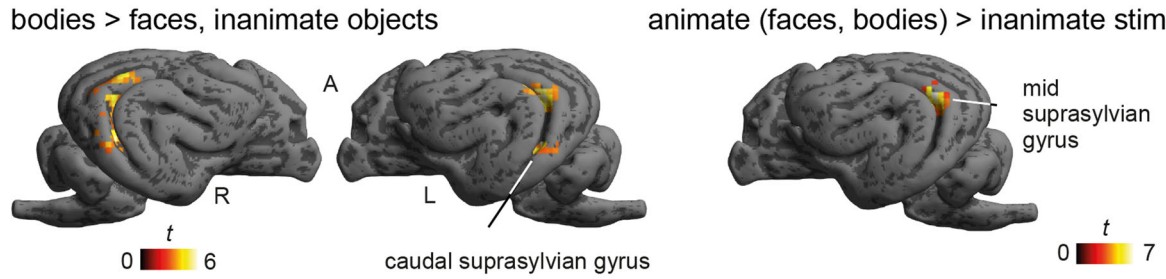

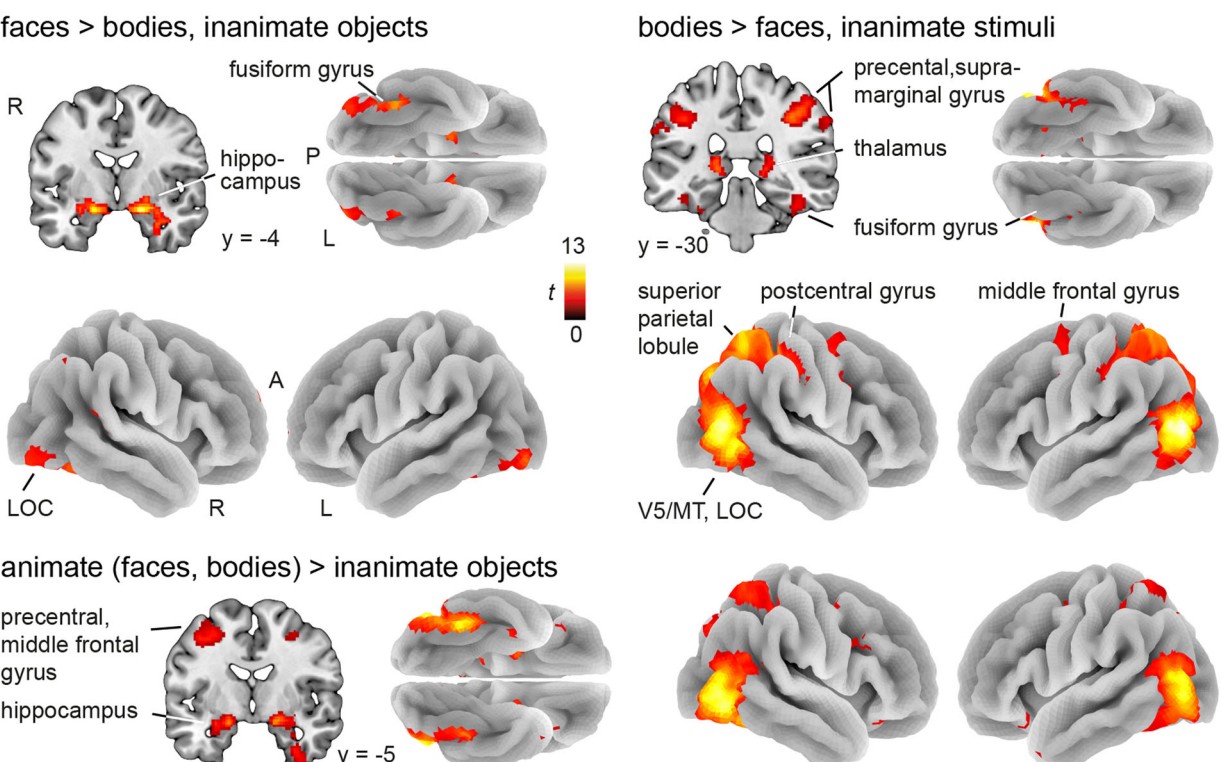

**Fig. 3 Complementary whole-brain univariate analysis confirms body- and animacy-sensitivity in both species, but face-sensitivity exclusive to humans. a** In dogs (*N* = 15), whole-brain univariate comparisons revealed body-sensitive regions (left, bodies > faces, inanimate objects) expanding across the mid and caudal suprasylvian gyrus. Both animate stimulus conditions (right, animate > inanimate stimuli) elicited activation in the mid suprasylvian gyrus. We did not observe face-sensitive regions in dogs (Supplementary Table S4). **b** In humans (*N* = 40), we found face-sensitive regions (left, faces > bodies, inanimate objects) in the lateral occipital cortex (LOC) and fusiform gyrus as well as activation in the hippocampus. Bodies compared to faces and inanimate objects also resulted in activation in the lateral occipital cortex and fusiform gyrus and additionally in the thalamus, the superior and the inferior parietal lobules, and the precentral and middle frontal gyrus (right, bodies > faces, inanimate objects). Overlapping occipito-temporal and fronto-parietal regions and the hippocampus also showed greater activation for faces and bodies (bottom, contrast animate > inanimate stimuli; Supplementary Table S5). All results are displayed at *p* < 0.05 FWE-corrected at cluster-level using a cluster-defining threshold of *p* < 0.001/0.005 for the human/dog data. R right, L left, A anterior, P posterior, MT middle temporal visual area (V5).

**Neural pattern similarities of animate compared to inanimate stimuli in higher-order visual areas of both species.** In the first part of our analysis, we focused on differences in mean activation levels to localize category-sensitive regions. However, even in the absence of specialized regions, differences in multivariate activation pattern similarities (i.e., neural representations) within and across regions might still be present between categories[10,37]. Therefore, we investigated the neural representations of faces and bodies in dogs and humans and their potential correspondence using whole-brain representational similarity analyses (RSA). We moved a 4 and 8 mm radius searchlight across the whole dog and

human brain, respectively, to determine individual pattern similarity maps between all trials (i.e., blocks) of each stimulus category[54,55]. We then conducted permutation-based paired *t*-tests to compare the pattern similarities maps of (1) animate vs. inanimate stimuli, (2) bodies vs. inanimate objects, (3) faces vs. inanimate objects, (4) faces vs. bodies, and (5, 6) within the face and body categories: images of conspecifics vs. heterospecifics (i.e., dog faces vs. human faces; dog bodies vs. human bodies) at the group-level.

Results revealed increased pattern similarity (i.e., higher correlation between all blocks of the same category) for animate

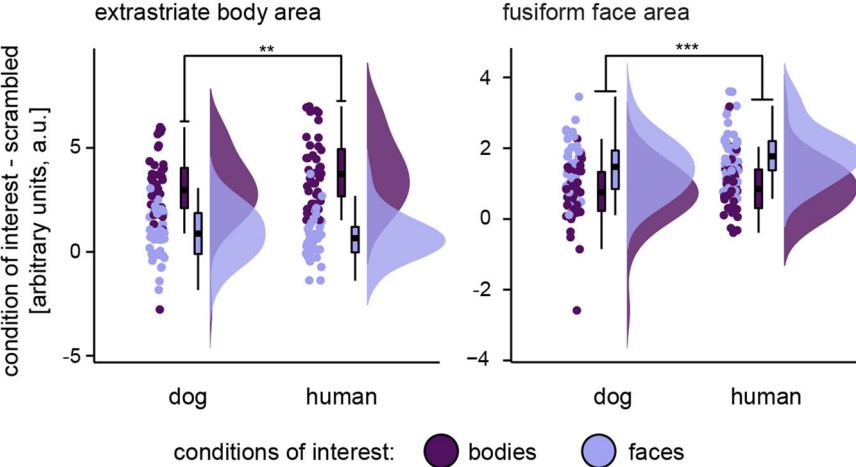

**Fig. 4 Preference for conspecifics in human extrastriate body and fusiform face area.** The human extrastriate body area showed greater activation levels for human (i.e., conspecific) bodies compared to dog bodies, and we found overall greater activation for human compared to dog stimuli (faces and bodies) in the human fusiform face area. Planned comparisons were false discovery rate (FDR) corrected to control for multiple comparisons ***$p < 0.001$, ****$p < 0.0001$. No such or other differences in fROI responses to con- and heterospecifics were found in dogs (all $p$'s > 0.1). Detailed information about the main effects of species (dog, human) and interaction between species and animate stimulus condition (face, body) can be found in Supplementary Table S6. Conditions of interest: bodies (purple), faces (lilac).

compared to inanimate stimuli in the occipito-temporal cortex of dogs (i.e., caudal suprasylvian) and humans (i.e., middle occipital gyrus) overlapping with the identified fROIs. This indicates different neural representations of animate compared to inanimate stimuli in higher-order visual regions of both species, corroborating the results of the univariate analyses. In human participants, we additionally observed increased similarity in the cerebellum and in fronto-parietal regions for animate compared to inanimate stimuli (Fig. 5); and increased pattern similarity for inanimate stimuli in the cerebellum (Supplementary Tables S9–S10).

**Increased pattern similarities in dog olfactory regions, but differences between faces and bodies exclusively in humans.** Moving on to neural representations for bodies and faces, we observed increased pattern similarity for bodies compared to inanimate objects in higher-order visual areas in the occipito-temporal cortex, partially overlapping with the identified fROIs, and the pattern similarity clusters expanded to the cerebellum in both species (Fig. 5a). Within the same regions (and again, in both species), we observed a higher correlation of activation patterns for conspecific compared to heterospecific bodies. Furthermore, when the dogs viewed dog compared to human bodies, results revealed increased similarity in clusters expanding across limbic structures and regions associated with olfaction (Fig. 5b). In humans, we observed increased similarity for human compared to dog bodies again in temporo-parietal regions (Fig. 5b). The reversed contrast (i.e., conspecific < heterospecific bodies) did not reveal significant pattern similarities in either species.

Seeing faces compared to inanimate objects resulted in significantly increased pattern similarity in occipito-temporal cortices of both species and, again, in dog olfactory structures (Fig. 5c). No significant pattern similarities for conspecific compared to heterospecific faces emerged, in neither species, but increased pattern similarity for dog compared to human faces in the bilateral human middle occipital gyrus and brainstem. In humans, we further observed a stronger correlation for inanimate objects compared to faces in the cerebellum, lingual and precentral gyrus; and in the inferior temporal gyrus compared to bodies (Supplementary Table S10).

Comparing pattern similarities for faces vs. bodies, we only found significant differences in the human sample. Observing faces compared to bodies elicited increased pattern similarity in the occipital lobe including the lingual gyrus and the neighboring cuneus and middle occipital gyrus (Fig. 5d). The reversed comparison revealed increased pattern similarity for bodies compared to faces in the occipito-temporal lobe including the middle occipital, lingual and fusiform gyrus, in fronto-parietal regions such as the superior parietal lobule and the orbitofrontal gyrus, as well as clusters including the insula and mid cingulate gyrus (Fig. 5e; see Supplementary Tables S9–S10 for detailed results and Fig. 6 for a schematic summary of the main results).

**Discussion**
Our findings show that in both humans and dogs the occipito-temporal cortex has a prominent role in the perception of animate entities. As predicted based on dogs' high-responsiveness to gestural and referential bodily cues[40–44], we found first evidence for a body-sensitive region (i.e., greater activation for bodies compared to faces and inanimate objects) in the dog temporal lobe. We further identified three occipito-temporal regions with a preference for faces and bodies compared to inanimate objects. By adding bodies as stimuli, and thus controlling for animacy, our findings crucially expand those from earlier investigations on face perception in dogs[33–37,39] and suggest that previously identified face-sensitive areas may respond more generally to animate entities. In humans, we replicated previous work localizing multiple distinct face- and body-sensitive regions (e.g., ref. [8], for review). Moreover, multivariate pattern analyses revealed neural representations (i.e., increased pattern similarity) of human and dog faces as well as dog bodies in dog olfactory areas. Importantly, in terms of their evolution, dogs and humans have split over 90 million years ago[56] and the neocortex of their last common ancestor consisted mainly of primary and secondary sensory regions[30]. Thus, higher-order sensory cortices of the two species cannot be considered homologous (see Fig. 6 for schematic summary of the results and comparison of the two brains). Hence, our findings suggest a convergent evolution[57] of the neural bases of animate vs.

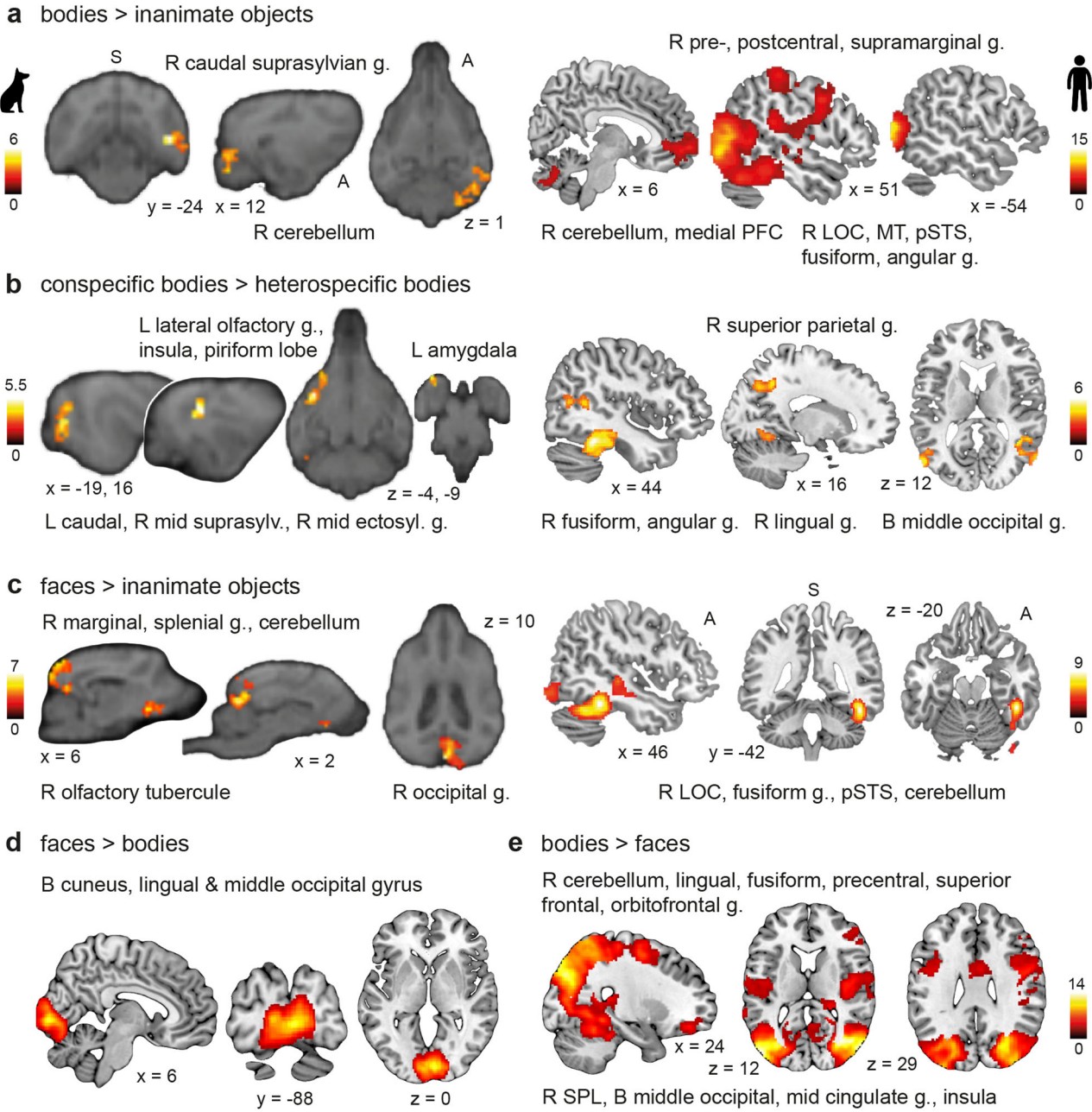

**Fig. 5 Neural pattern similarities of animate compared to inanimate stimuli in higher-order visual areas were increased in both species, while pattern similarities between faces and bodies differed only in humans. a, c** Show increased pattern similarities for bodies or faces compared to inanimate stimuli in higher-order visual areas and the cerebellum of dogs and humans. We also observed differences between the two species in regions beyond visual cortices. In dogs, as shown in **b, c**, we detected neural pattern similarities for faces and conspecific (=dog) bodies in clusters expanding across olfactory regions. **b** In humans, perceiving conspecific (=human) bodies revealed increased pattern similarity in fronto-parietal regions. Neural representations of faces vs. bodies only differed in humans, revealing (**d**) increased pattern similarities for faces compared to bodies in the posterior occipital lobe and (**e**) for bodies compared to faces in the lateral occipito-temporal lobe, insula, and fronto-parietal regions. All results are displayed at $p < 0.05$, FWE-corrected at cluster-level using a cluster-defining threshold of $p < 0.005/0.001$ for the dog/human data ($N = 15/40$; see also Supplementary Tables S9–10). Anatomical locations are shown in **b** for the dog and in d for the human data: superior (S), anterior (A); all sagittal, coronal, and axial planes displayed have the same orientation. Coordinates refer to a canine-breed averaged template[83] or to MNI space for the human data. t $t$-value (of paired $t$-tests); g. gyrus, PFC prefrontal cortex, LOC lateral occipital cortex, MT middle temporal visual area (V5), pSTS posterior superior temporal sulcus, SPL superior parietal lobule, R right, L left, B bilateral. The dog and human icons in panel a were purchased from thenounproject.com (royalty-free license).

inanimate stimuli perception but potential divergence regarding face and body perception in dogs and humans. The differential engagement of visual and olfactory brain functions fits particularly well with the differential sensitivity and preferential use of these perceptual systems to infer social and contextual information in humans and dogs.

Perceiving biological agents is crucial for survival and social relationships. Hence, the visual differentiation between animate vs. inanimate entities in both species reflects neural category representations of evolutionary importance. Eye-tracking studies show that dogs spontaneously look at images of human and dog faces, inanimate objects (i.e., toys) or alphabetical characters

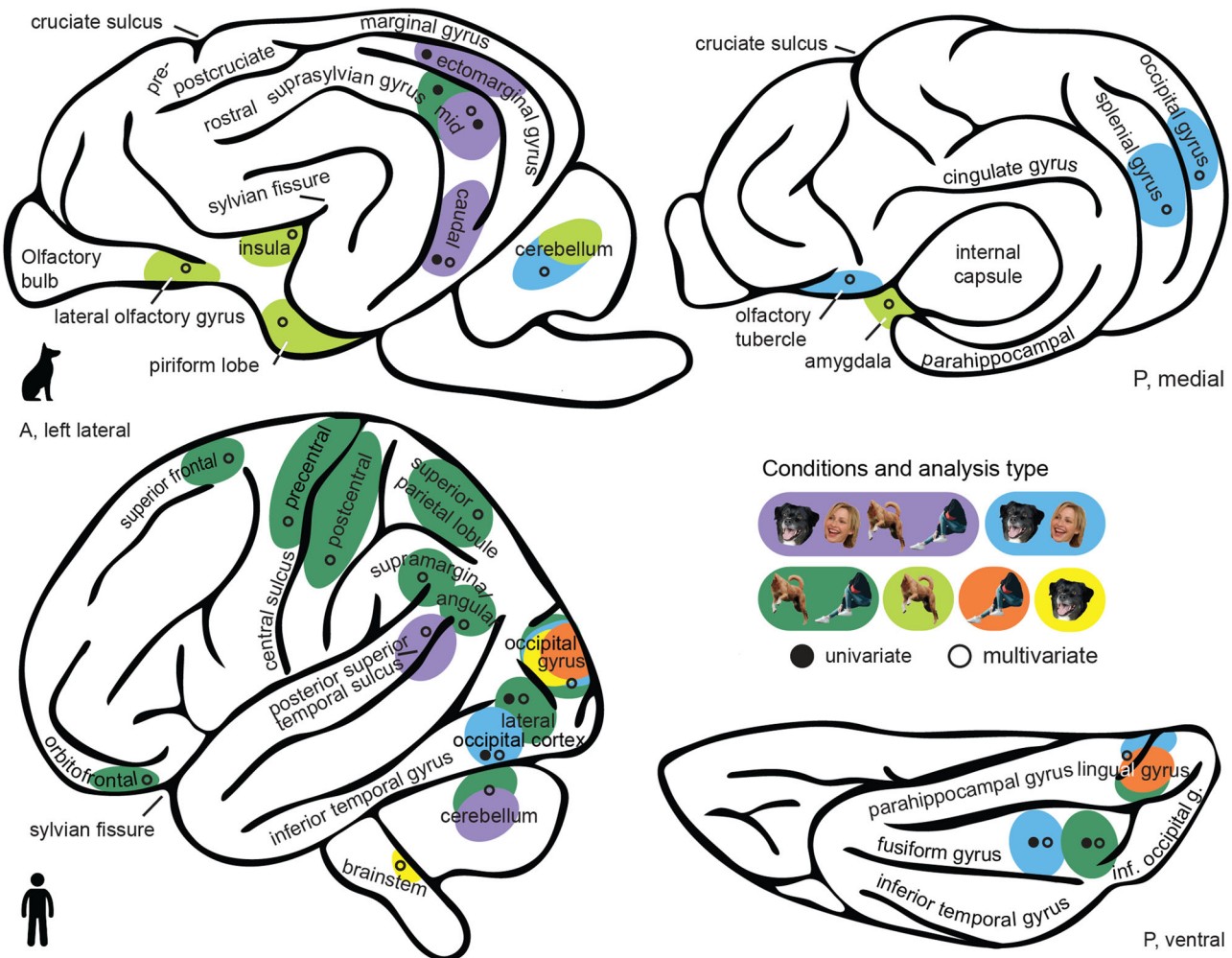

**Fig. 6 Graphical summary of the main study findings illustrating brain regions with analogous and divergent functions between both species.** The schematic brain figures show results from the functional regions-of-interest (fROIs univariate activation levels; black circle) and representational similarity analyses (RSA multivariate activation patterns; white circle). For visual guidance, we also labeled some anatomical landmarks, such as the cruciate (dog) and central sulcus (human), the parahippocampal and cingulate gyrus, as well as the (pseudo-)-sylvian fissure. For visual comparisons of the results, it is important to note that the last common ancestor of dogs and humans most likely had a smooth brain consisting mainly of primary and secondary sensory regions[30]; dog and human temporal lobes thus evolved independently and differ significantly in overall morphology[111,112]. The most significant landmark, the (pseudo-) sylvian fissure, is at the center of the dog temporal lobe with the gyri wrapped around but constitutes the border to the frontal- and parietal lobe in humans (see lateral views). To reduce complexity, observed results are always summarized on one hemisphere and they do not mark the exact but the approximate anatomical location. Also, increased pattern similarity for bodies compared to faces in the human mid cingulate gyrus and insula are not depicted. Example category images are license-free stock photos derived from www.pexels.com and were modified for the study purpose (i.e., head or body cut out). Conditions: faces and bodies of both species combined (purple), dog and human faces (blue), dog and human bodies (dark green), dog bodies (light green), human bodies (orange), dog faces (yellow). A anterior, P posterior. The dog and human icons were purchased from thenounproject.com (royalty-free license).

when presented on a screen and respond to novelty in all four categories, but fixate longer at faces compared to inanimate object or alphabetical characters[58]. They also target animate agents when, for example, presented with images of animals or humans embedded in natural landscapes[59] or of social interactions between humans and dogs[60]. Dividing stimuli into an animate vs. inanimate dimension is also one of the first visual categorizations formed by human infants[61]. In this way, animacy representation provides the first building block for more complex visual categorizations, such as faces vs. bodies. Importantly, we only observed the animacy-preference in higher-order visual and multimodal but not in primary visual areas of dogs and humans. This suggests that the results are unlikely to reflect mere differences in attention towards animate compared to inanimate stimuli (which should be visible already in primary visual

responses). Animacy might therefore constitute a general functional organizing principle in both mammalian brains, considering its biological significance and the observed cross-species similarities in our study.

Previous behavioral investigations of how dogs perceive bodies have mainly focused on the decoding of gestural cues[40–43] and dog neuroimaging studies so far have overlooked body perception entirely. We thus hope that localizing a novel region that preferentially processes non-facial bodily cues will inspire more research on how dogs perceive bodily social cues and if, for example, they are able to detect identity or emotional expressions equally well from bodies as they do from faces[17–19]. In the present study, we also localized several occipito-temporal regions that responded to animate stimuli more generally; this might further indicate that dogs, in comparison to humans, focus more on

whole-body social cues rather than on specific body parts. This interpretation is in line with a recent comparative eye-tracking study showing that dogs equally attend to a whole-body social cue (i.e., face and rest of the body), whereas humans spend significantly more time looking at the face[47]. Overall, our results do not contradict previous behavioral and imaging findings of dogs perceiving facial and bodily cues of dogs and humans[28,44,62–65] but might suggest that the majority of brain regions involved in the perception of faces are also involved in the perception of bodies.

A recent dog neuroimaging study[66] further demonstrated that the temporal regions observed in our study play a crucial part in the perception of complex social interactions, confirming their involvement in social perception and the pivotal role of the temporal lobe in social cognition. However, dogs might still have small additional body- or face-sensitive patches within the localized animacy-responsive regions that were not detectable using our non-invasive fMRI measures. For example, research in sheep (Ovis aries) has revealed 40 cells in a small patch of the temporal cortex preferentially responding towards faces using single-cell recordings[67]. Interestingly though, activation in most of the face-sensitive cells was modulated by dominance as indicated by the size of the horns, demonstrating that bodily information also plays an important role for social cognition in sheep.

Regarding other vs. own species perception, our results indicate greater activation for conspecific (i.e., human) compared to dog stimuli in half of the human face- and body-sensitive regions. Results regarding conspecific preference in human face processing regions have been mixed, reporting either greater[11,37,68,69] or comparable[16,70,71] activation levels for human compared to dog, macaque or other non-human animal faces. Preference for human compared to animal bodies has also been reported in previous work[72]. Overall, own-species preference appears to be more pronounced in non-human primates[12,16]. In dogs, we found no evidence for a preference for conspecifics in the occipito-temporal cortex of dogs. These results are in line with a previous dog fMRI study using a similar functional region-of-interest approach[34] but contrast with another study reporting conspecific-preferring regions in dog visual cortices[37]. Contrary to the latter study, we did not look for species-sensitive areas but asked if the localized face- and body-sensitive areas respond stronger to conspecifics compared to heterospecifics. No species-preferential processing within face- or body-sensitive regions also fits dogs' exceptional visuo-cognitive skills in encoding both dog and human facial and bodily cues. Previous behavioral and eye-tracking studies with dogs suggest no significant difference in the perception of human or dog positive emotional facial expressions[19], whole-body videos[47], or images of social interactions[60]. Moreover, two studies demonstrated that dogs, already as puppies, follow human gestural communication and show an interest in human faces[48], which was not observed in wolf puppies[73]. They could link variation in these socio-cognitive abilities to genetic factors, suggesting that dogs' attention to humans might be the result of selection pressure during domestication[74]. Dog cortical body- and animate-sensitive areas might therefore be tuned to respond equally to human and dog stimuli. However, we did find neural representations of conspecific compared to human bodies in dogs' limbic regions (i.e., amygdala, insula), which further emphasizes the need for behavioral research investigating the perception of emotional body cues in dogs. In the present study, we only used stimuli depicting neutral and positive displays. Prior behavioral and eye-tracking findings with dogs suggest an attentional bias towards angry or aggressive facial expressions of dogs[75] but an aversive effect of angry human facial expressions[18,58]. Future studies investigating conspecific preferences for faces or bodies in dogs should therefore consider adding negative emotional displays (of faces and bodies) and

heterospecific stimuli of other familiar species, such as cats, to investigate the effects of emotion and domestication on conspecific vs heterospecific perception.

In this context, it is important to note that in accordance with a potential divergent evolution of face, body, and species representations in dogs and humans we observed increased pattern similarity for faces (regardless of species) and conspecific (dog) bodies in dog higher-order olfactory association cortices using whole-brain representational similarity analysis. Faces compared to inanimate objects elicited increased pattern similarity in the olfactory tubercle which is situated posterior to the olfactory peduncle and anterior to the piriform lobe, and projects to the hypothalamus via the medial forebrain bundle[76]. Research in rodents suggests that this region plays an important role in the formation of odour preferences and reward processing[77], and that it might operate as a multi-sensory integration area (see refs. [78,79] for review). Conspecific compared to human bodies elicited increased pattern similarity in the piriform lobe expanding to the lateral olfactory gyrus (i.e., prepiriform paleocortex[76]). Research in humans suggests that the piriform cortex plays an important role in the encoding of odor representations (see e.g., ref. [80] for review). In dogs, the piriform cortex and olfactory bulb are active when they are presented with odors[80]. Both regions are connected with each other and the amygdala via an olfactory-piriform tract which also runs below the hippocampus[49] and the piriform cortex receives input from the ventral prefrontal cortex[81]. It also serves as a relay for a recently discovered large pathway connecting the olfactory bulb and visual cortex[49] where we also observed increased pattern similarity for faces compared to inanimate objects (i.e., marginal, splenial gyrus) and for conspecific compared to human bodies (i.e., clusters encompassing the caudal and mid suprasylvian, and mid ectosylvian gyrus). Overall, these findings reflect dogs' high olfactory sensitivity and its interplay with visual perception to infer social and contextual information[50].

Brain morphology systematically varies across breeds and correlates with behavioral specializations[82]. Most (86%) dogs in the present study were pure-bred herding or hunting breeds. Although our sample was homogenous regarding breeds, and all dogs had mesocephalic skull shapes, we opted for a functional region-of-interest (fROI) approach to account for slight deviations in activation peaks due to neuroanatomical variation and used a breed-averaged template[83] for whole-brain analyses. We have localized body- and animate-sensitive regions in the ectomarginal and suprasylvian gyrus. These regions have been identified as part of a network that systematically covaries in size (i.e., gray matter volume) across dog breeds and that is positively associated with behavioral specializations related to vision (e.g., herding or hunting), but also with explicit companionship[82], which further emphasizes the areas' role for social cognition. However, considering breed-specific neuroanatomical differences in olfactory and gustatory brain regions[82], the findings in dog olfactory cortices might be even more pronounced in dogs selectively bred for scent detection. In the present study, we did not have enough variance to test for potential differences between breeds; future studies and cumulative meta-analyses should, however, consider further investigating the link between behavioral specializations and the neural bases of face and body perception in the dog brain.

Our stimulus set was controlled for size, spatial extent, and luminance. We randomized image composition of stimuli blocks across participants and added a subset of scrambled versions of the images as visual control. We also conducted exploratory analyses comparing additional low-level visual properties measures and results revealed no differences in contrast across all stimuli but for hue and saturation. However, the differences in hue and saturation did not reflect the observed differences in

activation levels for bodies and faces. If the dog fROI results were driven by these low-level visual properties, we should have seen a differential response to human compared to dog bodies as they differ in hue. We also did not find differences in activation levels for dog and human faces although their saturation measures differ. Further, faces did not significantly differ from bodies and inanimate objects and vice versa for bodies compared to faces and inanimate objects in any of the low-level visual properties. Thus, they cannot explain face- or body-sensitivity. These findings are also in line with previous work, demonstrating that differences in hue or saturation do not explain face- or conspecific sensitivity in dogs and humans[37].

We also took several steps to maximize neural signal sensitivity already when designing our study (e.g., by using a block design) and at the analysis stage (e.g., by using dog-tailored haemodynamic models increasing response detection power[51]). However, the extensive training required for dogs[28] resulted in different sample sizes for the two species, and the more extensive prior work in humans[52] in more constrained search spaces. Nevertheless, we were still able to detect face and body-preferences in humans when we conducted the analysis again in 1000 randomly drawn human sub-samples (i.e., resampling analysis approach) with the identical fROI approach (i.e., anatomical mask) and sample size as for the dogs, indicating that the observed results were not driven by these methodological differences. However, considering that human and dog functional scans had the same image resolution, but the size of their brains significantly differs, it is, as mentioned earlier, possible that dogs do have small face-sensitive patches that were not detectable with the present setup. Research in humans has also shown that activation pattern similarities for visual categories such as faces or inanimate objects correlate with low-level visual properties[84], suggesting that visual features might serve as a stepping stone to form semantic representations of visual categories[85]. Since this aspect could not be addressed within our study design (i.e., randomized image composition, block design, controlled for certain low-level visual properties), future work needs to test if visual categories might be based on similar underlying visual features in dogs and humans, or if the observed differences in activation patterns between the two species might be explained based on different visual properties humans and dogs rely on to perceive visual stimuli. Overall, the present study marks the first step toward comparing body perception in the dog and human brain. However, more research is needed to elucidate further and compare the neural mechanisms underlying face and body perception in the dog and human brain.

In conclusion, our study reveals novel evidence for similarities and differences in animacy, face, and body perception between two phylogenetically distant mammal species, advancing our understanding of the foundations of social cognition and behavior and the evolution of the social brain. Finally, we provide first insights into the differentially evolved sensory systems of dogs and humans for the perception of faces and bodies.

## Methods

**Participants**. Fifteen trained[28], fully awake and unrestrained (Fig. 1a) family pet dogs (*Canis familiaris*; 11 females, age range: 4–11 ears, mean age: 7.8 years) consisting of 10 Border Collies, 2 Australian Shepherds, 1 Labrador Retriever and 2 mixed-breed dogs participated in the present study. We aimed to collect data from a minimum of $N = 12$ dogs during the data collection period from April 2019 to July 2020, which was the median sample size of task-based dog fMRI studies at the time of planning the study (the median sample size in 2022 was $N = 13.5$). All caregivers gave informed written consent to their dogs' participation. Dog data collection was approved by the institutional ethics and animal welfare commission in accordance with Good Scientific Practice (GSP) guidelines and national legislation at the University of Veterinary Medicine Vienna (ETK-06/06/2017), based on a pilot study conducted at the University of Vienna.

We collected comparative data from forty human participants (22 females, age range: 19–28 years, mean age: 23 years). We aimed for a sample size of $N = 40$

participants, based on previous studies in our lab with similar task designs, and also being in line with a previous study investigating the neural bases of face and body perception using a functional region-of-interest approach with a sample size of $N = 35$ participants reporting $F$-values above 40 for all repeated measures analyses[52]. Human participants were right-handed, had normal or corrected-to-normal vision, reported no history of neurological or psychiatric disease or phobia of dogs, fulfilled the standard inclusion criteria for functional MRI, and gave informed written consent. Human data collection was approved by the ethics committee of the University of Vienna (reference number: 00565) and performed in line with the latest revision of the Declaration of Helsinki (2013).

**Task and procedure**. We employed a block design (duration: 12 s) split in two 5 min runs, where participants saw images of faces and bodies of dogs or humans, inanimate objects, and scrambled versions of these images (5 images per block; Fig. 1b, Stimulus Material) on an MR-compatible screen (32 inch) positioned at the end of the scanner bore. Crucially, we used the same task for dogs and humans. Human participants were instructed to watch the images presented on the MR screen and dogs were trained to attend to the MR screen (passive viewing paradigm). Each run contained three blocks per condition and block order was randomized but the same condition was never presented twice in a row. Between blocks, participants saw a visual baseline jittered between 3–7 s with a white cross presented on gray background. Image composition for each block and order within each block was randomized across participants to ensure effects were not driven by specific blocks and each image was presented once.

**Stimulus material**. The stimulus set comprised 180 colored images of faces and bodies of dogs and humans, everyday inanimate objects (e.g., a toy, a chair), and phase-grid scrambled versions of each category (30 images per condition) derived from Wagner and colleagues[86], the Hemera Photo-Object database (Hemera Technologies) and the internet (see Fig. 1b for examples). In consultation with the dog trainers, we only selected images of inanimate objects dogs are familiar with in their everyday life. All images were resized to $600 \times 600$ pixels and presented in the center of the MR screen on gray background. In line with prior human and non-human primate neuroimaging studies (see e.g., refs. [11,16,72,87]), we edited out the heads, as well as objects (e.g., a coffee cup, a soccer ball) from the body images to disentangle body from face and object perception. To increase ecological validity, the face and body images showed a variety of postures (e.g., jumping, looking up), neutral and positive emotional displays (e.g., sleeping, smiling), and viewing perspectives (e.g., from above, from a side angle).

Luminance of all images was equalized across all images and backgrounds. Grid-scrambled images were created based on images equally drawn from each category (i.e., subset of each category) to control for potential low-level visual differences and complex visual stimulation. Face and body images vary in spatial extent (=ratio image/background) due to their shape (mean for faces = 58.53; for bodies = 29.42; $t(118) = 21.17$, $p < 0.0001$), matching them in spatial extent would have required resizing the face images to half their size resulting in less ecologically valid images, but we matched dog and human images within the body and face categories (means for faces: dogs = 58.55; humans = 58.50; $t(58) = 0.03$, $p = 0.98$; means for bodies: dogs = 30.97; humans = 27.87; $t(58) = 1.33$, $p = 0.19$). Further, half of the object images were matched in spatial extent to either body (mean for bodies = 29.42; for matched objects = 29.86, $t(73) = -0.17$, $p = 0.86$) or face (mean for faces = 58.53; for matched objects = 57.23, $t(73) = 0.78$, $p = 0.44$) images.

**Motion and attention**. During data collection, overall motion and wakefulness were live monitored via the camera of an eye-tracker (Eyelink 1000 Plus, SR Research, Ontario, Canada) positioned below the MR-compatible screen. For the dogs, we were able to see the entire head including the coil allowing trained staff to monitor their attention towards the screen throughout the data collection. The dog trainer stayed within the scanner room but out-of-sight throughout the scan session to monitor and handle the dogs. Human participants saw both task runs within a single scanner session with a short break in-between. For the dogs, the number of attempted sessions varied depending on how many repetitions they needed to complete one run without substantive motion and with sufficient attentiveness (i.e., eyes open and gazing towards the center of the screen); in-between task runs the dogs were always given a short break outside the MR scanner. As the visual scan path can affect activation levels in response to visual stimuli[88], we ensured that participants could see all stimuli equally well without having to perform frequent eye-movements. We used static stimuli, resized the images to $600 \times 600$ pixels and positioned them at the center of the MR screen to appear in the participant's eye-field. In addition, trained staff live monitored their gazing patterns and did not observe differences between stimulus categories. After each scan session we evaluated the motion parameters. If overall motion exceeded ≈ 4 mm (overall max. value: 4.2 mm) in any of the three translation directions, the dog re-invited to repeat the run in a subsequent session and sessions were scheduled at least one week apart (see also section MRI data preprocessing pipeline for further motion censoring applications). On average, dogs needed three sessions to complete both runs. No data of the non-successful sessions were used for analysis. Individual session numbers along with sample size descriptives are openly available on the Open Science Framework (osf.io/kzcs2).

**MRI data acquisition**. We acquired all MRI data with a 3 T Siemens Skyra MR-system (Siemens Medical, Erlangen, Germany) and a 15-channel coil (initially designed for measurements of the human knee) for data acquisition in dogs and a 32-channel human head coil for data acquisition in humans. Functional scans of dogs used a 2-fold multiband (MB) accelerated echo planar imaging (EPI) sequence including the following parameters: voxel size = $1.5 \times 1.5 \times 2$ mm$^3$, repetition time (TR) / echo time (TE) = 1000/38 ms, field of view (FoV) = $144 \times 144 \times 58$ mm$^3$, flip angle = 61°, 20% gap and 24 axial slices covering the whole brain (interleaved acquisition, descending order). On average, task runs consisted of 324 volumes, but numbers vary slightly due to manual stopping upon completion of the task-run (see https://osf.io/wefcz for individual volume numbers). Structural scans had a voxel size of 0.7 mm isotropic (TR/TE = 2100/3.13 ms, FoV = $230 \times 230 \times 165$ mm$^3$) and were acquired in a separate scan session prior to functional data collection. Human functional scans (on average: 271 volumes per run; individual volume numbers: https://osf.io/wefcz) were acquired using a 4-fold MB accelerated EPI sequence including the following parameters: voxel size = 2 mm isotropic, TE = 1200/34 ms, FoV = $192 \times 192 \times 124.8$ mm$^3$, flip angle = 66%, 20% gap and 52 axial slices coplanar to the connecting line between anterior and posterior commissure (interleaved acquisition, ascending order). Additionally, we obtained field map scans to correct functional scans for magnetic field inhomogeneities using a double echo gradient echo sequence with the following parameters: voxel size = $1.72 \times 1.72 \times 3.85$ mm$^3$, TR/TE1/TE2 = 400/4.92/7.38 ms, FoV = $220 \times 220 \times 138$ mm$^3$, flip angle = 60% and 36 axial slices (same orientation as functional scans). Structural scans had a voxel size of 0.8 mm isotropic (TR/TE = 2300/2.43 ms, FoV = $256 \times 256 \times 166$ mm$^3$) and were acquired after functional data acquisition.

**Data processing and statistical analysis**. Imaging data were pre-processed and analyzed using SPM12 (https://www.fil.ion.ucl.ac.uk/spm/software/spm12/), Matlab 2018b (MathWorks) and R 3.6.3[89].

*MRI data preprocessing.* In both samples, we slice-time corrected (reference: middle slice) and realigned functional images to the mean image. Human imaging data was also unwarped using the acquired field map. Dog imaging data was manually reoriented with the rostral commissure set as a visual reference (SPM module: "*Reorient images/Set origin*") to match the template orientation[83] and structural images were skull-stripped using individual binary brain masks created using itk-SNAP[90]. We co-registered the structural to the mean functional image, segmented the structural images in both samples and normalized the human data to the Montreal Neurological Institute (MNI) template space and the dog data to a breed-averaged stereotaxic template space[83]. Normalized images were resliced to 1.5 mm isotropic and smoothed with a 3-dimensional Gaussian kernel (full-width-at-half-maximum, FWHM; with twice the raw voxel resolution: dogs/humans = 3/4 mm; see ref. [51] for an in-depth description of our dog data preprocessing pipeline). We then calculated individual scan-to-scan motion (framewise displacement, FD) and added motion regressors to first-level general linear models (GLMs) for each scan exceeding the a priori set FD threshold of 0.5 mm (i.e., motion scrubbing[91,92]) to account for both translational and rotational displacements. For the dog participants, we removed on average 8% of the scans from each run (run 1: mean FD = 0.23 mm, 90th percentile = 0.36 mm; run 2: mean FD = 0.24 mm, 90th percentile = 0.38 mm) and 1% of the scans from each run of the human participants (run 1: mean FD = 0.17 mm, 90th percentile = 0.22 mm; run 2: mean FD = 0.18 mm, 90th percentile = 0.21 mm). Individual framewise displacement data and plots of the individual motion parameters (i.e., six realignment parameters) are openly available on the Open Science Framework (osf.io/kzcs2).

*Mass-univariate activation analysis.* We analysed the functional data using a GLM approach implemented in SPM12. Individual GLM matrices included six task regressors (dog faces, dog bodies, human faces, human bodies, inanimate objects, scrambled) and the six regressors from the realignment procedure along with the framewise displacement regressors were added as nuisance regressors. All blocks were estimated using a boxcar function time-locked to the onset of each block with a duration of 12 s. For the dog data, the signal was convolved with a tailored dog haemodynamic response function[51] (HRF), while the standard human canonical HRF (i.e., the default HRF parameters provided by SPM12) was used for the human data. The dog HRF reflects a previously observed 2–3 s earlier peak of the BOLD signal than expected by the human HRF model. Normalized, individual binary masks served as explicit masks, and we applied a high-pass filter with a cut-off at 128 s. First, we estimated contrast maps for faces, bodies, and inanimate objects (all conditions > scrambled control). We then split the data in two sets (task run 1, task run 2). Based on the data from the first task run, we estimated a visual stimulation contrast (all conditions > implicit visual baseline) to localize visual-responsive voxels and five subject-level contrast maps for the difference between our conditions of interest (i.e., faces, bodies with equal weights for human and dog images, objects) > scrambled and faces vs. bodies to define the functional regions of interest (fROIs). For the second task run, we computed seven subject-level contrasts, three for faces, bodies, and inanimate objects with each compared to scrambled controls for each task regressor and four differentiating between species and body part (i.e., dog bodies, human bodies, dog faces, human faces) compared to scrambled controls. From these contrasts, we extracted the parameter estimates for the fROI analysis.

*Functional region-of-interest approach.* We implemented a standard functional region-of-interest (fROI) approach to investigate potential category-specificity of cortical regions. The participant-level contrast images from the first run served as localizer data to define individual category-sensitive regions. Within anatomically constrained search spaces (see below and Fig. 2a) we first localized all voxels responding stronger to faces vs. bodies (i.e., face search spaces: faces > bodies, body search spaces: bodies > faces) and from these voxels we then selected the top-10% voxels from each hemisphere with the strongest signal for the condition-of-interest compared to inanimate objects (i.e., face areas: faces > objects, body areas: bodies >objects) to form bilateral individual fROIs. The data from the left-out second run allowed then to directly test potential category specificity in an independent data set. Thus, we extracted parameter estimates from the conditions-of-interest contrasted against the scrambled control from the individual body and face fROIs using the REX toolbox[93].

Choosing the top-10% voxels to define individual functional regions-of-interest (fROIs) was an a priori analytical decision we made based on the size of the fROIs before any activation levels were extracted. The aim was to create functional fROIs with a sufficient amount of voxels to be analyzed while still being able to detect potentially small category-sensitive regions in the dog brain. The chosen threshold resulted in mean fROI sizes ranging from 4.6 voxels (left occipital face fROIs) to 14.27 voxels (left splenial face fROI; see Supplementary Table S2 for all average fROI sizes and section Alternative top-% voxels threshold to define functional fROIs below).

*Anatomical search spaces.* We also localized face and body areas in restrained search spaces to retrieve anatomically more precise information. For the dog participants, we could not build on previous research due to different template spaces or data unavailability and therefore selected all task-responsive gyri as search spaces derived from a simple visual stimulation contrast (i.e., all conditions > implicit visual baseline, task run 1). Since the majority of significant clusters expanded across more than one anatomical region, we decided to not only select regions with a significant local maxima (Supplementary Table S3, Fig. 1c) but all gyri with visual-responsive voxels which were determined using the python software AtlasReader[94]. This resulted in six search spaces for potential face or body regions: mid suprasylvian, caudal suprasylvian, ectomarginal, occipital, marginal and splenial gyrus (Fig. 2a). For the human participants, we used bilateral fusiform and occipital face area parcels as face area search spaces and bilateral fusiform and extrastriate body area parcels as body area search spaces derived from previous research[52] (Fig. 2a). Not all parcels used in this study are mentioned in the paper but made openly available by the authors at https://web.mit.edu/bcs/nklab/GSS.shtml; we flipped (i.e., mirrored) right hemisphere parcels, if there were no left hemisphere parcels available.

*Group comparisons.* Group comparisons for the fROI analysis were performed running repeated measures analyses of variance (ANOVAs). First, we tested our main research question, whether the body areas resulted in increased sensitivity for bodies regardless of species and vice versa for faces compared to inanimate objects. Thus, we ran a one-way ANOVA with image category (faces, bodies, inanimate objects) as independent variable. Next, to investigate if there is a difference in activation between conspecific and heterospecific stimuli, we used $2 \times 2$ within-subjects ANOVAs (*species*: conspecific, heterospecific; image category: face, body). *P*-values for multiple planned-post hoc comparisons as well as for group comparisons investigating the same research questions were false-discovery rate (FDR) controlled. An example for the latter: For all six potential body fROIs we asked whether they result in greater activation levels for bodies compared to faces and inanimate objects.

For whole-brain univariate group analyses, we determined visual-responsive areas for the functional region-of-interest (fROI) analysis (see section anatomical search spaces above) performing a group-level activation comparison entering the visual stimulation contrast (all conditions > implicit visual baseline) from the first task run in a second-level one sample t-test. For a complementary whole-brain exploration of face and body perception, we conducted a one-way repeated measures analysis of variance (ANOVA; levels: faces, bodies, inanimate objects; all levels > scrambled controls) using the flexible factorial framework in SPM12 to explore face-, body- or animacy-sensitive regions (i.e., faces > bodies, inanimate objects; bodies > faces, inanimate objects; faces, bodes > inanimate objects). We determined significance on the group-level by applying cluster-level inference with a cluster-defining threshold of $p < 0.005/0.001$ (dogs/humans) and a cluster probability of $p < 0.05$ family-wise error (FWE) corrected for multiple comparisons. Cluster extent (i.e., minimum spatial extent to be labeled significant) was calculated using the SPM extension "CorrClusTh.m"[95].

*Alternative top-% voxels threshold to define functional fROIs.* We decided to use top-10% voxels as a threshold to create fROIs with sufficient data points that would still be able to detect potentially small category-selective regions in the dog brain. Thus, we decided based on the dog fROI sizes before any parameter estimates were extracted (see Supplementary Table S2 for mean fROI sizes). While choosing the top-10% was an a priori but analytical decision, we also performed validation analyses after completing the main analysis, using fROIs for the dog and human sample for percentage cut-offs ranging from 1% to 100% of the top active voxels in

steps of 5%, report parameter estimates for these percentages, and compare them to the main analysis with top-10% activated voxels.

*Sample size and search space differences.* In order to exclude the possibility that differences between dogs and humans were driven by methodological decisions or differences, we repeated the fROI analysis for the human participants using anatomical masks of the human fusiform gyrus[96] and calcarine sulcus including the surrounding cortex[97] to localize fROIs in the human sample (i.e., identical to the dogs). We then randomly resampled 1000 different sub-samples of $n = 15$ human participants (i.e., equal sample size as dogs) and conducted the same fROI analysis as in dogs with each sub-sample (i.e., resampling approach similar to multiverse analysis), as well as the identical analysis as in the main text for all $N = 40$ human participants including parameter estimates for different top-% cut-offs to define fROIs.

*Low-level visual properties.* The stimulus set was controlled for size, spatial extent, and luminance; and we added scrambled versions of a subset of these images as a control condition for visual stimulation and low-level visual properties (see Stimulus material for details). To test if observed activation levels might have been driven by underlying differences in other low-level visual properties, we conducted exploratory analyses to measure hue, saturation, and contrast of the stimulus material. First, we converted the images from RGB (red, green, blue) to HSV (hue, saturation, value/brightness) color space. Second, for each image, hue and saturation were calculated by measuring the mean value across all pixels for the respective components and contrast by computing the standard deviation of the pixel intensities[98] (i.e., value/brightness). Mirroring the functional region-of-interest analysis (fROI), we then conducted two ANOVAs for each of the low-level visual property measures to test for potential differences between stimuli categories. The first was a one-way ANOVA with image category (faces, bodies, inanimate objects) as independent variable and the second a two-way ANOVA with species (conspecific, heterospecific) and image category (face, body) as independent variables.

*Representational similarity analysis.* Next, we investigated the neural representations for faces, bodies and inanimate objects and their potential convergence in dogs and humans. To this end, we performed a whole-brain representational similarity analysis[54,55,99] (RSA) to determine neural pattern similarities within image categories. GLMs were modeled identical to univariate GLMs (see above) but for each block, we ran a separate GLM with the block as task regressor and remaining blocks were combined in one regressor of no interest[100]; runs were modeled independently. Thus, the analysis resulted in 36 single-trial beta estimates for each participant (6 conditions × 6 trials/blocks). RSA was performed using the smoothed functional data. For all RSA analyses, we moved a spherical searchlight (dogs: $r = 4$ mm, 81 voxels; humans: $r = 8$ mm, 251 voxel) throughout individual whole-brain gray matter masks computed based on the normalized segmentation output considering only searchlights with a minimum of 15 gray matter voxel for the dog and 30 for the human data.

For each participant, we extracted single-trial beta estimates from each voxel within a given searchlight. We then sorted them according to their stimulus category (dog/human bodies, dog / human faces, inanimate objects) and reshaped the data to a trial (i.e., blocks) × voxel matrix. Next, we computed a trial × trial similarity matrix by correlating values of each voxel from one single-trial beta estimate with the values of all other single-trial beta estimates applying Pearson correlation. Finally, in order to retrieve overall similarity scores, we then Fisher's $z$-transformed the data and calculated overall similarity matrices by averaging scores across the respective stimulus categories. We applied Fisher's $z$ transformation before averaging Pearson's $r$[101,102] to ensure normality and because it leads to a lower positive bias (i.e., overestimation) than the negative bias resulting from averaging non-transformed $r$[103]. We were specifically interested in pattern similarities across animate vs. inanimate (faces × bodies vs. in animate objects), faces or bodies vs. inanimate objects, faces vs. bodies and conspecific vs. heterospecific species dimensions within face and body categories. We then assigned the overall similarity values to the center voxel of each searchlight resulting in individual whole-brain pattern similarity maps.

At the group-level, we used permutation-based paired $t$-tests to compare the pattern similarities between trials of (a) faces vs. inanimate objects (i.e., [dog faces × human faces] vs. inanimate objects), (b) bodies vs. inanimate objects (i.e., [dog bodies × human bodies] vs. inanimate objects), (c) animate vs. inanimate images (i.e., [dog faces × human faces × dog bodies × human bodies] vs. inanimate objects), (d) faces vs. bodies (i.e., [dog faces × human faces] vs. [dog bodies × human bodies]), and within the face and body categories: images of conspecifics vs. heterospecifics (i.e., (e) dog faces vs. human faces; (f) dog bodies vs. human bodies). We computed permutation tests[104] to determine group-level significance on the cluster-level using the Statistical nonParametric Mapping (SnPm13, http://www.nisox.org/Software/SnPM13/) toolbox running 5000 permutations for each paired $t$-test and applied cluster-level inference with a cluster defining threshold of $p < 0.005/0.001$ (dogs/humans) and a cluster probability of $p < 0.05$ FWE corrected for multiple comparisons.

**Statistics and reproducibility.** We collected data from $N = 15$ pet dogs and $N = 40$ human participants. Data were analysed using SPM12 (https://www.fil.ion.ucl.ac.uk/spm/software/spm12/), Matlab 2018b (MathWorks) and R 3.6.3[89]. To create figures we mainly used the R packages ggplot2[105] and RainCloudPlots[106], and the python project nilearn (http://nilearn.github.io), as well as itk-SNAP[90] and MRIcron (https://www.nitrc.org/projects/mricron). The task was implemented using PsychoPy[107].

**Reporting summary**. Further information on research design is available in the Nature Portfolio Reporting Summary linked to this article.

## Data availability
Univariate and multivariate beta maps (Figs. 1, 3 and 5 and Supplementary Tables S1, 4, 5, 9, 10), individual raw functional region-of-interest (fROI) data (Figs. 2, 4, Supplementary Figs. S1–S3, Supplementary Tables S3, S6), low-level visual properties descriptives of the stimulus material (Supplementary Fig. S4, Supplementary Tables S7, S8), motion parameters and further sample descriptives (Supplementary Table S1) have been deposited at the Open Science Framework (OSF) and are publicly available at https://osf.io/kzcs2/[108]. Due to ethical constraints raw human neuroimaging data is made available upon request privacy. Raw dog neuroimaging data is publicly available at zenodo.org[109].

## Code availability
Custom R and Matlab code supporting this manuscript are available at the Open Science Framework[108] (https://osf.io/kzcs2/) and Github (https://github.com/magdalenaboch/fROI-analysis)[110].

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

## Acknowledgements

We thank Prof. Nancy Kanwisher for her helpful comments on the preliminary results and analysis plan. We also want to thank Morris Krainz, Anna Thallinger, Olaf Borghi, and Helena Manzenreiter for their help in collecting the data, Boryana Todorova for her support in preparing the stimulus set and all the dogs and their caregivers and human participants for taking part in this project. This project was supported by the Austrian Science Fund (FWF): W1262-B29 and by the Vienna Science and Technology Fund (WWTF) [10.47379/CS18012], the City of Vienna and ithuba Capital AG, and the Messerli Foundation (Sörenberg, Switzerland). The funders had no role in study design, data collection and analysis, decision to publish, or preparation of the manuscript.

## Author contributions

M.B.: conceptualization, methodology, software, validation, formal analysis, investigation, data curation, writing - original draft, writing - review & editing, visualization, project administration. I.C.W.: conceptualization, methodology, resources, writing - original draft, writing - review & editing, supervision. S.K.: investigation, writing - review & editing. L.H.: conceptualization, resources, writing - review & editing, supervision, funding acquisition. C.L.: conceptualization, methodology, resources, writing - original draft, writing - review & editing, supervision, funding acquisition.

## Competing interests

The authors declare no competing interests.
