## [Peer Review File · Communications Biology]

Reviewers' comments:

Reviewer #1 (Remarks to the Author):

This study investigated the neural correlates of face and body stimuli presented to dogs and humans using functional MRI. Six image categories were presented (headless dog bodies, headless human bodies, inanimate objects, dog faces, human faces, scrambled images) in a block design. Several analyses were conducted based on regions of interest and whole-brain data. The contrasts were appropriate. In brief, analyses revealed functional similarity in occipito-temporal cortex for passively viewing body stimuli across species. The findings involving dog olfactory areas are intriguing. Importantly, hue and saturation were controlled as potential causes of contrast differences. The introduction succinctly summarized the appropriate literature pointing out important limitations. The present study improves upon limitations in prior published findings. The findings are novel and will be important to readers interested in cognitive evolution and comparative neuroimaging, as they help reveal information about the evolution of social perception. Overall, the paper was well written. There are several things that need to be explained better (or perhaps I missed the detail).

1. I do not understand the logic of removing the heads from the body images. Could the authors please elaborate on why this is necessary to do? The headless bodies look strange and unfamiliar. Wouldn't this be a potential confound when comparing faces vs bodies in that dogs/humans are likely to have very little experience with headless bodies.

2. I think using a 10% arbitrary threshold is a weird choice. The authors argue why they made this choice in relation to other thresholds, but what is the necessity for making these arbitrary choices is not clearly explained. They mention the following in the legend of the supplementary figure where this choice is justified: "Smaller percentages would have been less sensitive for dog fROIs and increased fROI sizes would have been less sensitive in the human fusiform and occipital face area as illustrated by the overlapping 95% confidence intervals". This is similar to p-hacking and this type of logic is not acceptable in statistics. One should choose thresholds based on an underlying principle which you think is correct and you then interpret the results. Seeing the results and choosing the threshold is going the other way round. The authors also go on to show that whole brain analyses kind of supports the results they got from ROI analysis. Then why do the complicated ROI analyses with arbitrary thresholds is not clear. Is that a way to circumvent multiple comparisons corrections?

3. Fig S1 also shows that the problem with dog results is lower t-values and larger inter-individual variability. The human sub-sample analyses (Fig S2-S3) also have a relatively low t-value. So, lower sample size probably explains the lower t-value. But even with lower t-value, human subsample analysis has significantly smaller variance than dog results for a similar-sized sample. I don't know whether this variability is due to noise (movement), differences in acquisition parameters between dogs and humans, issues with dog compliance, or because in-plane resolution of dogs and humans were almost same, but the dog brain is much smaller (this may lead to blurring of effects in the relatively larger voxels of the dog). For these reasons, quantitative comparisons of results between the species and strong conclusions about them (in a biological sense) are problematic.

Reviewer #2 (Remarks to the Author):

The manuscript "Functionally analogous body- and animacy-responsive areas in the dog (*Canis familiaris*) and human occipito-temporal lobe" (by Boch, Wagner, Karl, Huber and Lamm) represents a well-written, thoughtful, and methodologically advanced piece of research, with decent sample sizes and intriguing findings that are important for the field. They show, for the first time, how human & dog brains differentiate faces and bodies from objects and low-level scrambled controls.

The methodology is well explained and carefully constructed. However, while the human brain imaging data often have specific ethical restrictions, the dog raw neuroimaging data could be openly shared to allow for reproducibility, and should be considered by the authors. The authors share the end-result data maps, but this is not usable for the replication of the analysis but only for checking if the results were reported reasonably.

I would ask the authors to engage in deeper reflection on how the results are discussed. In the abstract, the authors state that "...only humans had regions specialized for face perception". This is something one is unable to say on the basis of this kind of research; one can only state that the equivalent region was not found in dogs in the current study. It is always the uncomfortable way of the research that if you do not find something does not mean it is not there, only that it was not found. This is especially true for the paradigms used for the first time as in the current study; only through accumulating knowledge and similar findings elsewhere we can begin to collectively come to a conclusion.

More important is the reflection on the work in the field so far: how do the new findings fit into the picture so far, what are the possible reasons the face-selective region was not found in dogs? The authors carefully consider their stimulus low-level composition and sensitivity of the methodology, and note the fMRI result difference to the single-cell studies as conducted in sheep, but one could also consider e.g. the spatial differentiation in dog brains compared to those of humans, the effects of variation in the dog brain morphology, and perhaps the effects of dog breeds to the possible underlying functionality. This increases the usability of the current work and gives tools for the field to explore the issue further.

Also, for the discussion, instead of "faces are not specific for dogs" viewpoint, I suggest emphasizing "bodies may be equally specific as faces for dogs" viewpoint. This may better fit the literature and could be further studied in the future. Perhaps dogs can detect emotion, identity and such from faces, but maybe they are equally good with bodies. And while the faces are processed in a holistic manner in dogs as in humans, maybe bodies are also processed similarly. Or, as behavioral results of face processing are dense in many species (e.g. newly hatched chicks turn toward a face, Rosa-Salva et al. Dev Sci 2010) - do these face-specific processes already take place at subcortical level, perhaps also with dogs? Should this be targeted for further studies in the fMRI research?

A few minor comments are listed below:

line 66 ref. #20: Apart from other references given here, this is a review instead of original research, so could be omitted here (or mentioned as such).

lines 81-82: "prior work did not find greater activation for faces compared to scrambled images" -> strictly speaking, this is untrue, as Kujala et al (Sci Rep 2020) did find a difference between scrambled and intact dog/human facial images in dog brain responses. However, this study also lacked bodies as a control stimulus, and the information retrieved with a neurophysiological method taps partially different neural mechanisms from the fMRI, in which the quicker responses may be

difficult to detect.

lines 107-111: Pls add that you conducted the fROI analysis in the individual level. This is a very important specification as ROIs are determined in a number of different ways to overcome the multiple comparison problem, and it already answers some further questions.

lines 116-135: This paragraph already describes the main findings. As it is highly unusual for the introduction, I suggest omitting unless the journal instructions specifically instructed the authors to include findings in the introduction.

Figure 6: The text in the figure is very small (actually, the same applies to other figures as well) and colors remain very difficult to distinguish from another, which is crucial for this figure. Please change one of the blues and the darker green color for the figure to be more readable (also in the CMYK print form, which people still do use). In the dog brain image, the color actually appears similar as the faces-only color in the label – better distinction in the colors would be beneficial here.

lines 387-390: About the conspecific vs. non-conspecific preference for human brain, original results showing this kind of differentiation could be referred to (e.g. Blonder et al Cogn Brain Res 2004).

lines 397-399: "Previous behavioural studies with dogs suggest no significant difference in the perception of human or dog emotional facial expressions"
-> This is not true; Somppi et al (Plos One 2016) did find a differential response for aggressive human & dog facial expressions in dogs, with implications of learned processing of human aggressive faces; these results are in line with the findings of Mueller et al (Curr Biol 2015). There are not many studies so far about this particular issue, but the reason this kind of difference is not often found may be in the differences of the stimulus salience; if emotional stimuli are too mild, differences are not found. In the current data, the emotionality appeared very low; perhaps the emotionality of the stimuli adds to the species-specific processing via innate & learned processing.

lines 410-412: Pls add the analysis which gave this result. With different analyses in the same context, it is difficult to follow which analysis is discussed.

lines 429-430: "Note also that" -> pls consider "Notably" instead

lines 433-434: "It is important though that we did not predict these findings and should therefore be regarded as preliminary." -> I do not see this necessary in this context. If the multiple comparisons problem is properly dealt with, a result is a result – it is not dependent on the psychic abilities of the researchers. Of course, it is appropriate to point out the need to examine this further and to replicate elsewhere.

Reviewer #1 (Remarks to the Author):

This study investigated the neural correlates of face and body stimuli presented to dogs and humans using functional MRI. Six image categories were presented (headless dog bodies, headless human bodies, inanimate objects, dog faces, human faces, scrambled images) in a block design. Several analyses were conducted based on regions of interest and whole-brain data. The contrasts were appropriate. In brief, analyses revealed functional similarity in occipito-temporal cortex for passively viewing body stimuli across species. The findings involving dog olfactory areas are intriguing. Importantly, hue and saturation were controlled as potential causes of contrast differences. The introduction succinctly summarized the appropriate literature pointing out important limitations. The present study improves upon limitations in prior published findings. The findings are novel and will be important to readers interested in cognitive evolution and comparative neuroimaging, as they help reveal information about the evolution of social perception. Overall, the paper was well written. There are several things that need to be explained better (or perhaps I missed the detail).

Author response:

We thank the reviewer for the very positive evaluation of our work, the constructive comments towards improving it further, and for confirming our view that this is novel and important work.

1. I do not understand the logic of removing the heads from the body images. Could the authors please elaborate on why this is necessary to do? The headless bodies look strange and unfamiliar. Wouldn't this be a potential confound when comparing faces vs bodies in that dogs/humans are likely to have very little experience with headless bodies.

We understand the reviewer's concern and would like to explain our rationale. Our goal was to disentangle category sensitivity for faces vs. bodies. By removing the faces from the body images, we attempted to achieve a balance between the naturalistic depiction of bodies and experimental control. We agree with the reviewer that dogs only have little exposure to headless bodies. However, dogs typically also do not see isolated faces (without bodies). Several behavioural studies still showed that isolated faces can be used to investigate familiarity, emotion perception, or emotion discrimination in dogs¹⁻³. Additionally, similar stimuli (isolated faces, isolated bodies) have been used in studies with humans and non-human primates (see e.g.,⁴⁻⁷), successfully showing differential neural representations for faces vs. bodies.

We now explain this in more detail in our updated manuscript (see below for relevant text passages).

Material and methods, lines 544-549: *"In line with prior human and non-human primate neuroimaging studies (see e.g.,⁴⁻⁷), we edited out the heads, as well as objects (e.g., a coffee cup, a soccer ball) from the body images to disentangle body from face and object perception. To increase ecological validity the face and body images showed a variety of postures (e.g., jumping, looking up), neutral and positive emotional displays (e.g., sleeping, smiling), and viewing perspectives (e.g., from above, from a side angle)."*

2. I think using a 10% arbitrary threshold is a weird choice. The authors argue why they made this choice in relation to other thresholds, but what is the necessity for making these arbitrary choices is not clearly explained. They mention the following in the legend of the supplementary figure where this choice is justified: "Smaller percentages would have been less sensitive for dog fROIs and increased fROI sizes would have been less sensitive in the human fusiform and occipital face area as illustrated by the overlapping 95% confidence intervals". This is similar to

p-hacking and this type of logic is not acceptable in statistics. One should choose thresholds based on an underlying principle which you think is correct and you then interpret the results. Seeing the results and choosing the threshold is going the other way round. The authors also go on to show that whole brain analyses kind of supports the results they got from ROI analysis. Then why do the complicated ROI analyses with arbitrary thresholds is not clear. Is that a way to circumvent multiple comparisons corrections?

We thank the reviewer for pointing out that further clarification is needed. We structured our reply in several parts.

(1) Rationale for conducting the functional region-of-interest (fROI) and exploratory whole-brain analyses

The main aim of the univariate analyses was to investigate whether dogs and humans have comparable face- and body-sensitive brain regions. We applied a functional region-of-interest (fROI) analysis approach to address this research question (see e.g.,⁸ for recent application to investigate category-sensitivity in human infants) and split the data into two independent data sets: (a) a localizer data set (first task run) to define individual potential face- or body-sensitive areas in visual-responsive brain regions and (b) a test data set (second task run) to extract activation levels from these regions

We chose this approach for two main advantages. First, defining individual fROIs within constrained search spaces accounts for slight variations of activation peaks between participants, as reported in past dog neuroimaging studies^{35,37}. Second, this approach allowed us to directly test the category-sensitivity of the localized category-sensitive regions using the left-out data set. This analysis approach has also been used in a recent human infant fMRI study to investigate face and body-sensitivity⁸.

While we considered this the ideal approach, fROI analyses have their known limitations, such as being “blind” to areas outside the fROIs tested. This is why we performed a complementary and exploratory whole-brain analysis. The results largely confirmed the fROI findings, but as expected showed lower sensitivity and detected only one animate-sensitive area, in the mid suprasylvian gyrus. Importantly, it revealed no areas additional to those focused on in the fROI analyses. Taken together, this complementary approach bolsters our interpretations regarding category-sensitive areas in the dog or human brain.

Regarding the reviewer’s concern that this analysis approach may have been “a way to circumvent multiple corrections,” **we emphasize that all analyses, i.e., both the fROI analyses and the whole-brain analyses were corrected for multiple comparisons, as explicitly stated in the manuscript**, on page 8 (lines 125-126) and throughout the methods and results sections.

(2) Rationale for choosing the top-10% most active voxels as a threshold to form functional regions-of-interest (fROIs)

This comment seems to suggest that we engaged in questionable research practices. This is not the case – we did not conduct *p*-hacking or any related practices, and we fully agree with the reviewer that “peaking” at data and then selecting a threshold is highly problematic. This is why choosing the top-10% most active voxels was an *a priori* analytical decision that we made solely based on the size of the resulting fROIs, *before* we extracted any parameter estimates and analysed them. This approach aimed to strike a balance between fROIs that would contain a sufficient amount of voxels to be analyzed, while still being able to detect potentially small category-selective regions in the dog brain.

After conducting the main fROI analysis using this threshold, however, we received feedback on this analysis approach at conferences and lab talks that suggested evaluating our decision to choose this 10% threshold using supplementary exploratory analysis (see **Supplementary Figure S1**, next page). Note that this is also evident by the fact that the first version of this study's preprint did not contain the supplementary analysis (see <https://www.biorxiv.org/content/10.1101/2021.08.17.456623v1?versioned=true>).

The rationale behind the supplementary analysis was to collect additional information on how much different thresholds may affect the results. This should also provide information for future dog neuroimaging studies. We, therefore, extracted parameter estimates for percentage cut-offs ranging from 1% to 100%. The results showed that smaller fROI sizes were less category-sensitive in all search spaces in the dog brain (see **Supplementary Figure S1**, next page). In contrast, in the human fusiform face area and occipital face area, thresholds above 60% resulted in fROIs that could not detect differences in activation levels between faces and bodies. Thus, the human fROIs likely also contained voxels that are not face- or body-sensitive at this size. For the dogs, larger fROI sizes resulted in comparable outcomes in the mid and caudal suprasylvian gyrus but had less sensitivity in the ectomarginal gyrus. Taken together, the supplementary analysis confirmed, in a qualitative fashion, that the top-10% voxel threshold indeed strikes an optimal balance in the definition of category-sensitive fROIs.

The reviewer's comment made it evident that our rationale and approach require a more elaborate explanation. We thus carefully revised all relevant sections in the manuscript as follows:

Results, lines 135-139: "We chose this approach **for two main advantages**. First, defining individual fROIs accounted for differences in the location of activation peaks between participants (as reported in previous studies^{35,37}). Second, this allowed **us** to not only localize potential face- or body-sensitive regions but also to directly evaluate their category-sensitivity using the left-out data."

Results, lines 168-175: "Choosing the top 10% voxels was an a priori analytical decision **we made based on the size of the resulting individual fROIs before any activation levels were extracted (see Materials and Methods: Functional region-of-interest approach for details)**. However, after we conducted the main analysis, we also extracted parameter estimates for a range of different percentage cut-offs between 1% to 100% **to validate the results using this threshold**, altogether confirming that the 10% threshold was an appropriate fROI size for detecting relevant activation levels (see **Supplementary Note 1 and Supplementary Figure S1**)."

Results, lines 213-215: "Next, we conducted an exploratory whole-brain analysis to complement the fROI analysis **and investigate if we can detect the category-sensitive areas using whole-brain group comparisons**."

Materials and Methods, Functional region-of-interest approach, lines 680-687: "Choosing the top 10% voxels to define individual functional regions-of-interest (fROIs) was an a priori analytical decision we made based on the size of the fROIs before any activation levels were extracted. The aim was to create functional fROIs with a sufficient amount of voxels to be analyzed while still being able to detect potentially small category-sensitive regions in the dog brain. The chosen threshold resulted in mean fROI sizes ranging from 4.6 voxels (left occipital face fROIs) to 14.27 voxels (left splenial face fROI; see **Supplementary Table S2 for all average fROI sizes and section Alternative top-% voxels threshold to define functional fROIs below**)."

Materials and Methods, Alternative top-% voxels threshold to define functional fROIs, lines 731-740: "We decided to use top-10% voxels as a threshold to create fROIs with sufficient data points that would still be able to detect potentially small category-selective regions **in the dog brain**. Thus, we decided based on the dog fROI sizes before any parameter estimates were extracted (see **Supplementary Table S2 for mean fROI sizes**). While choosing the top-10% was an a priori but analytical decision, we also performed validation analyses **after completing the main analysis**, using fROIs for the dog and human sample for percentage cut-offs ranging from 1% to 100% of the top active

voxels in steps of 5%, report parameter estimates for these percentages, and compare them to the main analysis with top-10% activated voxels.”

Supplementary Note 1, Validation of threshold to define individual functional regions-of-interest:

“Individual functional regions-of-interest (fROIs) were determined based on the top-10% most active voxels within each anatomical search space. To validate the chosen threshold and investigate, for example, if smaller fROI sizes would have been able to detect face-sensitive areas in the dog brain, we extracted parameter estimates for percentage cut-offs ranging from 1% to 100%. The results show that smaller fROI sizes were less sensitive in all anatomical search spaces in the dog brain (see **Supplementary Figure S1**). In the human fusiform face area and occipital face area, thresholds above 60% resulted in fROIs that could not detect differences in activation levels between faces and bodies. Thus, the fROIs likely also contained voxels that are not face- or body-sensitive at this size. For the dogs, larger fROI sizes revealed comparable outcomes in the mid and caudal suprasylvian gyrus but had less sensitivity in the ectomarginal gyrus than lower thresholds. Thus, comparing fROI-defining thresholds confirmed 10% as a sufficient threshold to define category-sensitive fROIs in the dog and human brain.”

Activation levels for functional regions of interest (fROIs) based on varying top-% voxels thresholds

Supplementary Figure S1. Exploratory analysis of parameter estimates for faces, bodies, and inanimate objects retrieved from individual functional regions of interest (fROIs) defined based on top-% most active voxels for faces or bodies > inanimate objects (run 1) ranging from 1% to 100% in steps of 5%. For dogs, fROIs defined based on thresholds below 10% were less sensitive in all anatomical search spaces. Large fROI-defining thresholds were less sensitive in the human fusiform and occipital face area and dog ectomarginal gyrus as illustrated by the overlapping 95% confidence intervals (CIs). Points represent the mean. a.u., arbitrary units. The dashed line represents the a priori selected 10% threshold used for the main analysis that we set out to validate.

3. Fig S1 also shows that the problem with dog results is lower t-values and larger inter-individual variability. The human sub-sample analyses (Fig S2-S3) also have a relatively low t-value. So, lower sample size probably explains the lower t-value. But even with lower t-value, human subsample analysis has significantly smaller variance than dog results for a similar-sized sample. I don't know whether this variability is due to noise (movement), differences in acquisition

parameters between dogs and humans, issues with dog compliance, or because in-plane resolution of dogs and humans were almost same, but the dog brain is much smaller (this may lead to blurring of effects in the relatively larger voxels of the dog). For these reasons, quantitative comparisons of results between the species and strong conclusions about them (in a biological sense) are problematic.

We fully agree with the reviewer and therefore discuss the possibility of face-sensitive areas in the dog brain that were simply not detectable with our setup in our manuscript. We now also explicitly mention the difference in brain sizes between the two species but the same image resolution as a methodological limitation. Furthermore, we carefully revised the interpretation of our results in the manuscript. We now refrain from strong conclusions and emphasize that more studies and meta-analyses are needed to further elucidate our understanding of face and body perception in the dog brain. For example, we exchanged phrasings like “only humans have a specialized region for face perception” with “we only detected face-sensitive regions in humans”.

Abstract, lines 6-9: “Combining univariate and multivariate analysis approaches, we found functionally analogous occipito-temporal regions involved in the perception of animate entities and bodies in both species **and face-sensitive regions in humans.**”

Introduction, lines 99-104: “The analysis also revealed analogous occipito-temporal brain areas sensitive for animate entities (i.e., faces or bodies) compared to inanimate objects and low-level visual controls in dogs and humans indicating a convergent evolution of these neural bases. However, **we only detected** face-sensitive areas **in humans.** This suggests that previously identified face-responsive areas in the dog brain **may** respond more generally to animate compared to inanimate stimuli.”

Discussion, lines 311-314: “By adding bodies as stimuli, and thus controlling for animacy, our findings crucially expand those from earlier investigations on face perception in dogs⁹⁻¹⁴ and suggest that previously identified face-sensitive areas **may** respond more generally to animate entities.”

Discussion, lines 322-324: “Hence, our findings suggest a convergent evolution¹⁵ of the neural bases of animate vs. inanimate stimuli perception but **potential** divergence regarding face and body perception in dogs and humans.”

Discussion, lines 352-355: “**In the present study, we also localized several occipito-temporal regions that responded to animate stimuli more generally; this** might further indicate that dogs, in comparison to humans, focus more on whole-body social cues rather than on specific **body** parts.”

Discussion, lines 471-478: “Nevertheless, we were still able to detect face and body-preferences in humans when we conducted the analysis again in 1000 randomly drawn human sub-samples (i.e., resampling analysis approach) with the identical fROI approach (i.e., anatomical mask) and sample size as for the dogs, indicating that the observed results were not driven by these methodological differences. **However, considering that human and dog functional scans had the same image resolution, but the size of their brains significantly differs, it is, as mentioned earlier, possible that dogs do have small face-sensitive patches that were not detectable with the present setup.**”

Discussion, lines 487-490: “**Overall, the present study marks the first step towards comparing body perception in the dog and human brain. However, more research is needed to elucidate further and compare the neural mechanisms underlying face and body perception in the dog and human brain.**”

Reviewer #2 (Remarks to the Author):

The manuscript “Functionally analogous body- and animacy-responsive areas in the dog (*Canis familiaris*) and human occipito-temporal lobe” (by Boch, Wagner, Karl, Huber and Lamm) represents a well-written, thoughtful, and methodologically advanced piece of research, with decent sample sizes and intriguing findings that are important for the field. They show, for the first time, how human & dog brains differentiate faces and bodies from objects and low-level scrambled controls.

Author response:

We thank the reviewer for this overall very positive evaluation of our work and the constructive suggestions to improve the interpretation and reproducibility of our findings.

1. The methodology is well explained and carefully constructed. However, while the human brain imaging data often have specific ethical restrictions, the dog raw neuroimaging data could be openly shared to allow for reproducibility and should be considered by the authors. The authors share the end-result data maps, but this is not usable for the replication of the analysis but only for checking if the results were reported reasonably.

We thank the reviewer for this great suggestion. The dog neuroimaging data is now publicly available at zenodo.org¹⁶. Due to ethical reasons, we are indeed not able to share the raw human neuroimaging data but we will share them with researchers upon request. We now explicitly mention this option in our updated manuscript.

To improve the reproducibility of our analysis, we carefully explained all analysis steps and the rationale behind them in the manuscript and made the analysis scripts and raw functional region-of-interest (fROI) data publicly available. We also share group beta maps to enable future meta-analyses.

Revised sections in the manuscript:

Data availability, lines 825-827: “Univariate and multivariate beta maps, individual raw functional region-of-interest (fROI) data, motion parameters, low-level visual properties descriptives of the stimulus material and further sample descriptives have been deposited at the Open Science Framework (OSF) and are publicly available at <https://osf.io/kzcs2/>. Raw human neuroimaging data is made available upon reasonable request and raw dog neuroimaging data is publicly available at zenodo.org¹⁶.”

2. I would ask the authors to engage in deeper reflection on how the results are discussed. In the abstract, the authors state that “...only humans had regions specialized for face perception”. This is something one is unable to say on the basis of this kind of research; one can only state that the equivalent region was not found in dogs in the current study. It is always the uncomfortable way of the research that if you do not find something does not mean it is not there, only that it was not found. This is especially true for the paradigms used for the first time as in the current study; only through accumulating knowledge and similar findings elsewhere we can begin to collectively come to a conclusion.

We fully agree with the reviewer and have carefully revised the interpretation of our results in the manuscript. We now refrain from strong conclusions and emphasize that more studies and meta-analyses are needed to further elucidate our understanding of face and body perception in the dog brain.

Abstract, lines 6-9: “Combining univariate and multivariate analysis approaches, we found functionally analogous occipito-temporal regions involved in the perception of animate entities and bodies in both species and face-sensitive regions in humans.”

Introduction, lines 99-104: “The analysis also revealed analogous occipito-temporal brain areas sensitive for animate entities (i.e., faces or bodies) compared to inanimate objects and low-level visual controls in dogs and humans indicating a convergent evolution of these neural bases. However, **we only detected** face-sensitive areas **in humans**. This suggests that previously identified face-responsive areas in the dog brain **may** respond more generally to animate compared to inanimate stimuli.”

Discussion, lines 311-314: “By adding bodies as stimuli, and thus controlling for animacy, our findings crucially expand those from earlier investigations on face perception in dogs⁹⁻¹⁴, and suggest that previously identified face-sensitive areas **may** respond more generally to animate entities.”

Discussion, lines 322-324: “Hence, our findings suggest a convergent evolution¹⁵ of the neural bases of animate vs. inanimate stimuli perception but **potential** divergence regarding face and body perception in dogs and humans.”

Discussion, lines 352-355: “**In the present study, we also localized several occipito-temporal regions that responded to animate stimuli more generally; this might further indicate that dogs, in comparison to humans, focus more on whole-body social cues rather than on specific body parts.**”

Discussion, lines 487-490: “**Overall, the present study marks the first step towards comparing body perception in the dog and human brain. However, more research is needed to elucidate further and compare the neural mechanisms underlying face and body perception in the dog and human brain.**”

3. More important is the reflection on the work in the field so far: how do the new findings fit into the picture so far, what are the possible reasons the face-selective region was not found in dogs? The authors carefully consider their stimulus low-level composition and sensitivity of the methodology, and note the fMRI result difference to the single-cell studies as conducted in sheep, but one could also consider e.g. the spatial differentiation in dog brains compared to those of humans, the effects of variation in the dog brain morphology, and perhaps the effects of dog breeds to the possible underlying functionality. This increases the usability of the current work and gives tools for the field to explore the issue further.

We thank the reviewer for raising these important points. We structured our reply into several parts.

Spatial resolution

As mentioned by the reviewer, we already critically discussed methodological differences that might explain why we did not find a face-sensitive region in the dog brain. In terms of spatial resolution, it is of course also important to note that the brain sizes of dogs and humans significantly differ. Thus, considering that functional scans of both species had the same image resolution, sensitivity to detect small face-sensitive regions was lower for dogs compared to humans. We have now added this limitation to the discussion of our manuscript.

Discussion, lines 471-478: “**Nevertheless, we were still able to detect face and body-preferences in humans when we conducted the analysis again in 1000 randomly drawn human sub-samples (i.e., resampling analysis approach) with the identical fROI approach (i.e., anatomical mask) and sample size as for the dogs, indicating that the observed results were not driven by these methodological differences. However, considering that human and dog functional scans had the same image resolution, but the size of their brains significantly differs, it is, as mentioned earlier, possible that dogs do have small face-sensitive patches that were not detectable with the present setup.**”

Variation of brain morphology and breed differences

Brain morphology systematically varies across dog breeds¹⁷. Our sample of pet dogs was rather homogenous in terms of dog breeds (80% Border collies) and behavioural specializations, and all dogs had mesocephalic skull shapes. We further chose an analysis

approach for the main univariate analysis that accounts for slight deviations of activation peaks, e.g., due to neuroanatomical variations (i.e., functional region-of-interest analysis) and used a breed-averaged template¹⁸ to improve image registration for whole-brain analyses.

Our findings indicate body- and animate-sensitive regions in the dog suprasylvian and ectomarginal gyrus. Both regions have been identified as part of a network that systematically covaries in size (i.e., grey matter volume) across dog breeds¹⁷ and that is positively associated with behavioural specializations related to vision (herding, sight hunting) but also with explicit companionship, which further emphasizes the role of these areas for social cognition. Considering reported breed-specific differences of grey matter volume in olfactory and gustatory regions¹⁷, the observed involvement of dog olfactory cortices in face and body perception in our study might be more pronounced in breeds selectively bred for olfactory skills such as scent hunting or detection.

In the present study, we did not have enough variance to test for potential differences between breeds; future studies, and cumulative meta-analyses should, however, consider further investigating the link between behavioural specializations and the neural bases of face and body perception in the dog brain.

We have now added this important discussion to our manuscript and hope that it will inspire exciting new research into the effects of dog breeds and behavioural specializations.

Discussion, lines 433-450: *“Brain morphology systematically varies across breeds and correlates with behavioural specializations¹⁷. Most (86%) dogs in the present study were pure-bred herding or hunting breeds. Although our sample was homogenous regarding breeds, and all dogs had mesocephalic skull shapes, we opted for a functional region-of-interest (fROI) approach to account for slight deviations in activation peaks due to neuroanatomical variation and used a breed-averaged template¹⁸ for whole-brain analyses. We have localized body- and animate-sensitive regions in the ectomarginal and suprasylvian gyrus. These regions have been identified as part of a network that systematically covaries in size (i.e., grey matter volume) across dog breeds and that is positively associated with behavioural specializations related to vision (e.g., herding or hunting), but also with explicit companionship¹⁷, which further emphasizes the areas’ role for social cognition. However, considering breed-specific neuroanatomical differences in olfactory and gustatory brain regions¹⁷, the findings in dog olfactory cortices might be even more pronounced in dogs selectively bred for scent detection. In the present study, we did not have enough variance to test for potential differences between breeds; future studies and cumulative meta-analyses should, thus, consider further investigating the link between behavioural specializations and the neural bases of face and body perception in the dog brain.”*

4. Also, for the discussion, instead of “faces are not specific for dogs” viewpoint, I suggest emphasizing “bodies may be equally specific as faces for dogs” viewpoint. This may better fit the literature and could be further studied in the future. Perhaps dogs can detect emotion, identity and such from faces, but maybe they are equally good with bodies. And while the faces are processed in a holistic manner in dogs as in humans, maybe bodies are also processed similarly. Or, as behavioral results of face processing are dense in many species (e.g. newly hatched chicks turn toward a face, Rosa-Salva et al. Dev Sci 2010) - do these face-specific processes already take place at subcortical level, perhaps also with dogs? Should this be targeted for further studies in the fMRI research?

Face and body perception

We agree with the viewpoint that bodies may be equally important or specific as faces for dogs. We have therefore argued that our findings might indicate that dogs focus more on whole-body social cues rather than specific body parts and that our results do not contradict previous behavioural findings of dogs perceiving facial cues of conspecifics and humans¹⁹⁻²⁴ but might suggest that the majority of brain regions involved in the perception of faces are also involved in the perception of bodies. We have now carefully revised the discussion of our findings to emphasize this interpretation further and added suggestions for future studies

to investigate, for example, if dogs are able to detect identity or emotional expressions equally well from bodily stimuli as they do from faces^{1,2,25}.

Sub-cortical involvement

The exploratory whole-brain univariate analysis did not indicate any involvement of sub-cortical brain areas in dogs during face or body perception. However, representational similarity analysis revealed increased pattern similarity for conspecific compared to human bodies in the amygdala and insula cortex. These findings further emphasize the need for future investigations of emotion and species perception in dogs using whole-body or bodily stimuli. We have added this important insight to the discussion of our manuscript.

Discussion, lines 347-355: *“Previous behavioural investigations of how dogs perceive bodies have mainly focused on the decoding of gestural cues²⁶⁻²⁹ and dog neuroimaging studies so far have overlooked body perception entirely. We thus hope that localizing a novel region that preferentially processes non-facial bodily cues will inspire more research on how dogs perceive bodily social cues and if, for example, they are able to detect identity or emotional expressions equally well from bodily stimuli as they do from faces^{1,2,25}. In the present study, we also localized several occipito-temporal regions that responded to animate stimuli more generally, this might further indicate that dogs, in comparison to humans, focus more on whole-body social cues rather than on specific body parts. This interpretation is in line with a recent comparative eye-tracking study showing that dogs equally attend to a whole-body social cue (i.e., face and rest of the body), whereas humans spend significantly more time looking at the face³⁰. Thus, our results do not contradict previous behavioural and imaging findings of dogs perceiving facial and bodily cues of dogs and humans¹⁹⁻²⁴ but might suggest that the majority of brain regions involved in the perception of faces are also involved in the perception of bodies.”*

Discussion, lines 396-401: *“Dog cortical body- and animate-sensitive areas might therefore be tuned to respond equally to human and dog stimuli. However, we did find neural representations of conspecific compared to human bodies in dogs’ limbic regions (i.e., amygdala, insula), which further emphasizes the need for behavioural research investigating the perception of emotional body cues in dogs.”*

A few minor comments are listed below:

5. line 66 ref. #20: Apart from other references given here, this is a review instead of original research, so could be omitted here (or mentioned as such).

We removed the reference.

6. lines 81-82: “prior work did not find greater activation for faces compared to scrambled images”

-> strictly speaking, this is untrue, as Kujala et al (Sci Rep 2020) did find a difference between scrambled and intact dog/human facial images in dog brain responses. However, this study also lacked bodies as a control stimulus, and the information retrieved with a neurophysiological method taps partially different neural mechanisms from the fMRI, in which the quicker responses may be difficult to detect.

The focus of this paragraph was on fMRI studies, but we agree with the reviewer that the EEG study by Kujala et al (2020) should also be mentioned to have a better overview on non-invasive neuroimaging research investigating face perception in dogs and therefore added the study to the introduction.

Introduction, lines 60-67: *“Apart from one electroencephalography (EEG) study³¹, prior neuroimaging studies did not find greater activation for faces compared to scrambled images^{10,13}, but compared to scenes¹⁰ or objects^{9,10,14}, or didn’t have any non-facial controls¹¹, questioning if face-sensitivity rather reflects differences in low-level visual properties. Further, almost all prior studies*

lacked animate stimuli other than faces^{9-11,13,14,31} and the only study¹² with another animate stimulus category (i.e., the back of the head) had no inanimate control condition.”

7. lines 107-111: Pls add that you conducted the fROI analysis in the individual level. This is a very important specification as ROIs are determined in a number of different ways to overcome the multiple comparison problem, and it already answers some further questions.

We added this information to the introduction.

Introduction, lines 88-92: *“For our main analysis, we employed a functional region of interest (fROI) analysis on the individual level to investigate face- and body-sensitivity in the occipito-temporal cortex of dogs and humans and whether these regions responded differently to conspecific vs. heterospecific stimuli, as indicated by differences in activation levels.”*

8. lines 116-135: This paragraph already describes the main findings. As it is highly unusual for the introduction, I suggest omitting unless the journal instructions specifically instructed the authors to include findings in the introduction.

This is a journal specific requirement. As stated in the style and formatting guide of communications biology (<https://www.nature.com/documents/commsj-life-style-formatting-guide-accept.pdf>), the final paragraph of the introduction should be a brief summary of the major results and conclusions.

9. Figure 6: The text in the figure is very small (actually, the same applies to other figures as well) and colors remain very difficult to distinguish from another, which is crucial for this figure. Please change one of the blues and the darker green color for the figure to be more readable (also in the CMYK print form, which people still do use). In the dog brain image, the color actually appears similar as the faces-only color in the label – better distinction in the colors would be beneficial here.

We changed colours and made sure they are distinguishable in CMYK and RGB print form and increased the font size (see below). We also carefully revised the font size of all other figures to improve readability.

Figure 6. Graphical summary of the main study findings illustrating brain regions with analogous and divergent functions between both species. The schematic brain figures show results from the functional regions-of-interest (fROIs; univariate activation levels) and representational similarity analyses (RSA; multivariate activation patterns). For visual guidance, we also labelled some anatomical landmarks, such as the cruciate (dog) and central sulcus (human), the parahippocampal and cingulate gyrus, as well as the (pseudo)-sylvian fissure. For visual comparisons of the results, it is important to note that the last common ancestor of dogs and humans most likely had a smooth brain consisting mainly of primary and secondary sensory regions³²; dog and human temporal lobes thus evolved independently and differ significantly in overall morphology^{33,34}. The most significant landmark, the (pseudo-) sylvian fissure, is at the centre of the dog temporal lobe with the gyri wrapped around but constitutes the border to the frontal- and parietal lobe in humans (see lateral views). To reduce complexity, observed results are always summarized on one hemisphere and they do not mark the exact but the approximate anatomical location. Also, increased pattern similarity for bodies compared to faces in the human mid cingulate gyrus and insula are not depicted. Example category images are license-free stock photos derived from www.pexels.com and were modified for the study purpose (i.e., head or body cut out).

10. lines 387-390: About the conspecific vs. non-conspecific preference for human brain, original results showing this kind of differentiation could be referred to (e.g. Blonder et al Cogn Brain Res 2004).

We thank the reviewer for pointing out the missing discussion of this finding. Own-species (i.e., conspecific) preference appears to be more pronounced in non-human primates compared to humans^{7,35}. This might also explain the mixed results in previous human fMRI studies, reporting either greater^{5,12,36,37} or comparable^{7,38,39} activation levels for human compared to dog, macaque or other non-human animal faces. In regard to conspecific

preference in the extrastriate body area, we replicated previous results⁴⁰. We added this information and relevant references to the discussion of our manuscript.

Discussion, lines 374-382: “Regarding other vs. own species perception, our results indicate greater activation for conspecific (i.e., human) compared to dog stimuli in half of the human face- and body-sensitive regions. Results regarding conspecific preference in human face processing regions have been mixed, reporting either greater^{5,12,36,37} or comparable^{7,38,39} activation levels for human compared to dog, macaque or other non-human animal faces. Preference for human compared to animal bodies has also been reported in previous work⁴⁰. Overall own-species preference appears to be more pronounced in non-human primates^{7,35}. In dogs, we found no evidence for a preference for conspecifics in the occipito-temporal cortex of dogs.”

11. lines 397-399: “Previous behavioural studies with dogs suggest no significant difference in the perception of human or dog emotional facial expressions” -> This is not true; Somppi et al (Plos One 2016) did find a differential response for aggressive human & dog facial expressions in dogs, with implications of learned processing of human aggressive faces; these results are in line with the findings of Mueller et al (Curr Biol 2015). There are not many studies so far about this particular issue, but the reason this kind of difference is not often found may be in the differences of the stimulus salience; if emotional stimuli are too mild, differences are not found. In the current data, the emotionality appeared very low; perhaps the emotionality of the stimuli adds to the species-specific processing via innate & learned processing.

We thank the reviewer for pointing out this important detail in our discussion of the results.

Using eye-tracking, Somppi et al.⁴¹ investigated dogs’ viewing patterns towards human and dog emotional facial expressions. The study showed that dogs only looked longer at dog compared to human faces if the dog stimuli depicted threatening, aggressive expressions. The dogs also exhibited a greater attentional bias towards threatening dog faces but avoidance of threatening human faces. In the study from Müller et al.¹, dogs were presented with pairs of happy and angry human facial expressions. Half of the dogs received a reward for choosing the negative emotional display, and the other half for choosing the positive one. While all dogs could discriminate the happy and angry facial emotional expressions, dogs rewarded for selecting the negative display learned this association slower. This further emphasizes that dogs perceive angry human facial expressions as aversive stimuli.

Thus, although dogs are able to discriminate human and dog emotional facial expressions², findings from Somppi et al.⁴¹ and Müller et al.¹ show species-specific differences for the perception of negative facial expressions. This might result in differential activation levels for dog compared to human angry emotional expressions. Future neuroimaging studies investigating conspecific preferences or species perception should therefore consider using both positive and negative emotional displays.

For the present study, we only selected images with neutral or positive emotional displays (as described in **4.2.1 Stimulus material**) and did not find any differences in activation for human or dog stimuli in dog body- or animate-sensitive areas which is also well in line with the findings from Somppi et al.⁴¹.

We added this information and relevant references to the discussion of our results.

Discussion, lines 398-408: “Previous behavioural and eye-tracking studies with dogs suggest no significant difference in the perception of human or dog positive emotional facial expressions², whole-body videos³⁰, or images of social interactions⁴². Moreover, two studies demonstrated that dogs, already as puppies, follow human gestural communication and show an interest in human faces⁴³, which was not observed in wolf puppies⁴⁴. They could link variation in these socio-cognitive abilities to genetic factors, suggesting that dogs’ attention to humans might be the result of selection

pressure during domestication⁴⁵. Dog cortical body- and animate-sensitive areas might therefore be tuned to respond equally to human and dog stimuli. However, we did find neural representations of conspecific compared to human bodies in dogs' limbic regions (i.e., amygdala, insula), which further emphasizes the need for behavioural research investigating the perception of emotional body cues in dogs. In the present study, we only used stimuli depicting neutral and positive displays. Prior behavioural and eye-tracking findings with dogs suggest an attentional bias towards angry or aggressive facial expressions of dogs⁴¹ but an aversive effect of angry human facial expressions^{1,46}. Future studies investigating conspecific preferences for faces or bodies in dogs should therefore consider adding negative emotional displays (of faces and bodies) and heterospecific stimuli of other familiar species, such as cats, to investigate the effects of emotion and domestication on conspecific vs heterospecific perception."

12. lines 410-412: Pls add the analysis which gave this result. With different analyses in the same context, it is difficult to follow which analysis is discussed.

We were referring to the whole-brain representational similarity analysis. We added this information to the text.

Discussion, lines 409-413: In this context, it is important to note that in accordance with a potential divergent evolution of face, body, and species representations in dogs and humans we also observed increased pattern similarity for faces (regardless of species) and conspecific (dog) bodies in dog higher-order olfactory association cortices using whole-brain representational similarity analysis.

13. lines 429-430: "Note also that" -> pls consider "Notably" instead

We rephrased this section based on the reviewer's comment #3 and the word "notably" was therefore no longer fitting.

Discussion, lines 444-446: "However, considering breed-specific neuroanatomical differences in olfactory and gustatory brain regions¹⁷, the present findings in dog olfactory cortices might be even more pronounced in dogs selectively bred for scent detection."

14. lines 433-434: "It is important though that we did not predict these findings and should therefore be regarded as preliminary." -> I do not see this necessary in this context. If the multiple comparisons problem is properly dealt with, a result is a result – it is not dependent on the psychic abilities of the researchers. Of course, it is appropriate to point out the need to examine this further and to replicate elsewhere.

We removed the sentence from the discussion.

Discussion, lines 430-432: "Overall, these findings reflect dogs' high olfactory sensitivity and its interplay with visual perception to infer social and contextual information⁴⁷."

References

1. Müller, C. A., Schmitt, K., Barber, A. L. A. & Huber, L. Dogs can discriminate emotional expressions of human faces. *Curr. Biol.* **25**, 601–605 (2015).
2. Albuquerque, N. *et al.* Dogs recognize dog and human emotions. *Biol. Lett.* **12**, 20150883 (2016).
3. Somppi, S., Törnqvist, H., Hänninen, L., Krause, C. M. & Vainio, O. How dogs scan familiar and inverted faces: An eye movement study. *Anim. Cogn.* (2014) doi:10.1007/s10071-013-0713-0.
4. Downing, P. E., Chan, A. W.-Y., Peelen, M. V., Dodds, C. M. & Kanwisher, N. Domain specificity in visual cortex. *Cereb. Cortex* **16**, 1453–1461 (2006).
5. Tsao, D. Y., Moeller, S. & Freiwald, W. A. Comparing face patch systems in macaques and humans. *Proc. Natl. Acad. Sci. U. S. A.* **105**, 19514–19519 (2008).
6. Moeller, S., Freiwald, W. A. & Tsao, D. Y. Patches with links: A unified system for processing faces in the macaque temporal lobe. *Science* **320**, 1355–1359 (2008).
7. Tsao, D. Y., Freiwald, W. A., Knutsen, T. A., Mandeville, J. B. & Tootell, R. B. H. Faces and objects in macaque cerebral cortex. *Nat. Neurosci.* **6**, 989–995 (2003).
8. Kosakowski, H. *et al.* Selective responses to faces, scenes, and bodies in the ventral visual pathway of infants. *Curr. Biol.* **32**, 265–274.e5 (2022).
9. Cuaya, L. V., Hernández-Pérez, R. & Concha, L. Our Faces in the Dog's Brain: Functional Imaging Reveals Temporal Cortex Activation during Perception of Human Faces. *PLoS One* **11**, e0149431 (2016).
10. Dilks, D. D. *et al.* Awake fMRI reveals a specialized region in dog temporal cortex for face processing. *PeerJ* **3**, 3:e11115 (2015).
11. Thompkins, A. M. *et al.* Separate brain areas for processing human and dog faces as revealed by awake fMRI in dogs (*Canis familiaris*). *Learn. Behav.* **46**, 561–573 (2018).
12. Bunford, N. *et al.* Comparative Brain Imaging Reveals Analogous and Divergent Patterns of Species and Face Sensitivity in Humans and Dogs. *J. Neurosci.* **40**, 8396–8408 (2020).
13. Szabó, D. *et al.* On the Face of It: No Differential Sensitivity to Internal Facial Features in the Dog Brain. *Front. Behav. Neurosci.* **14**, 25 (2020).
14. Gillette, K. D., Phillips, E. M., Dilks, D. D. & Berns, G. S. Using Live and Video Stimuli to Localize Face and Object Processing Regions of the Canine Brain. *Animals* **12**, (2022).
15. Roberts, R. J. V., Pop, S. & Prieto-Godino, L. L. Evolution of central neural circuits: state of the art and perspectives. *Nat. Rev. Neurosci.* 2022 1–19 (2022) doi:10.1038/s41583-022-00644-y.
16. Boch, M., Wagner, I. C., Karl, S., Huber, L. & Lamm, C. Data from: Functionally analogous body- and animacy-responsive areas in the dog (*Canis familiaris*) and human occipito-temporal lobe [Data set]. *Zenodo* (2023) doi:10.5281/zenodo.7691966.
17. Hecht, E. E. *et al.* Significant Neuroanatomical Variation Among Domestic Dog Breeds. *J. Neurosci.* **39**, 7748–7758 (2019).
18. Nitzsche, B. *et al.* A stereotaxic breed-averaged, symmetric T2w canine brain atlas including detailed morphological and volumetrical data sets. *Neuroimage* **187**, 93–103 (2019).
19. Buttelmann, D. & Tomasello, M. Can domestic dogs (*Canis familiaris*) use referential emotional expressions to locate hidden food? *Anim. Cogn.* **16**, 137–145 (2013).
20. Barber, A. L. A., Randi, D., Müller, C. A. & Huber, L. The Processing of Human Emotional Faces by Pet and Lab Dogs: Evidence for Lateralization and Experience Effects. *PLoS One* **11**, e0152393 (2016).
21. Karl, S., Boch, M., Virányi, Z., Lamm, C. & Huber, L. Training pet dogs for eye-tracking and awake fMRI. *Behav. Res. Methods* **52**, (2020).
22. Thompkins, A. M. *et al.* Dog–human social relationship: representation of human face familiarity and emotions in the dog brain. *Anim. Cogn.* **24**, 251–266 (2021).
23. Nagasawa, M., Murai, K., Mogi, K. & Kikusui, T. Dogs can discriminate human smiling faces from blank expressions. *Anim. Cogn.* **14**, 525–533 (2011).
24. Siniscalchi, M., Lusito, R., Vallortigara, G. & Quaranta, A. Seeing Left- or Right-Asymmetric Tail Wagging Produces Different Emotional Responses in Dogs. *Curr. Biol.* **23**, 2279–2282

- (2013).
25. Huber, L., Racca, A., Scaf, B., Virányi, Z. & Range, F. Discrimination of familiar human faces in dogs (*Canis familiaris*). *Learn. Motiv.* **44**, 258–269 (2013).
 26. Duranton, C., Range, F. & Virányi, Z. Do pet dogs (*Canis familiaris*) follow ostensive and non-ostensive human gaze to distant space and to objects? *R. Soc. Open Sci.* **4**, 170349 (2017).
 27. Range, F., Virányi, Z. & Huber, L. Selective Imitation in Domestic Dogs. *Curr. Biol.* **17**, 868–872 (2007).
 28. Soproni, K., Miklósi, Á., Topál, J. & Csányi, V. Dogs' (*Canis familiaris*) Responsiveness to Human Pointing Gestures. *J. Comp. Psychol.* **116**, 27–34 (2002).
 29. Topál, J., Kis, A. & Oláh, K. Dogs' Sensitivity to Human Ostensive Cues: A Unique Adaptation? in *The Social Dog: Behavior and Cognition* (eds. Kaminski, J. & Marshall-Pescini, S.) 319–346 (Elsevier, 2014). doi:10.1016/B978-0-12-407818-5.00011-5.
 30. Correia-Caeiro, C., Guo, K. & Mills, D. Bodily emotional expressions are a primary source of information for dogs, but not for humans. *Anim. Cogn.* **24**, 267–279 (2021).
 31. Kujala, M. V. *et al.* Time-resolved classification of dog brain signals reveals early processing of faces, species and emotion. *Sci. Reports 2020 101* **10**, 1–13 (2020).
 32. Kaas, J. H. Reconstructing the Areal Organization of the Neocortex of the First Mammals. *Brain. Behav. Evol.* **78**, 7–21 (2011).
 33. Bryant, K. L. & Preuss, T. M. A Comparative Perspective on the Human Temporal Lobe. *Digit. Endocasts* 239–258 (2018) doi:10.1007/978-4-431-56582-6_16.
 34. Lyras, G., Geer, A. van der & Dermitzakis, M. Evolution of the brain of Plio/Pleistocene wolves. *Cranium* **18**, 30–40 (2001).
 35. Hori, Y. *et al.* Interspecies activation correlations reveal functional correspondences between marmoset and human brain areas. *Proc. Natl. Acad. Sci. U. S. A.* **118**, e2110980118 (2021).
 36. Kanwisher, N., Stanley, D. & Harris, A. The fusiform face area is selective for faces not animals. *Neuroreport* **10**, 183–187 (1999).
 37. Chao, L. L., Martin, A. & Haxby, J. V. Are face-responsive regions selective only for faces? *Neuroreport* **10**, 2945–2950 (1999).
 38. Blonder, L. X. *et al.* Regional brain response to faces of humans and dogs. *Cogn. Brain Res.* **20**, 384–394 (2004).
 39. Tong, F., Nakayama, K., Moscovitch, M., Weinrib, O. & Kanwisher, N. RESPONSE PROPERTIES OF THE HUMAN FUSIFORM FACE AREA. *Cogn. Neuropsychol.* **17**, 257–280 (2010).
 40. Downing, P. E., Chan, A. W. Y., Peelen, M. V., Dodds, C. M. & Kanwisher, N. Domain Specificity in Visual Cortex. *Cereb. Cortex* **16**, 1453–1461 (2006).
 41. Somppi, S. *et al.* Dogs evaluate threatening facial expressions by their biological validity - Evidence from gazing patterns. *PLoS One* **11**, (2016).
 42. Törnqvist, H. *et al.* Comparison of dogs and humans in visual scanning of social interaction. *R. Soc. Open Sci.* **2**, 150341 (2015).
 43. Bray, E. E. *et al.* Early-emerging and highly heritable sensitivity to human communication in dogs. *Curr. Biol.* **31**, 1–5 (2021).
 44. Salomons, H. *et al.* Cooperative Communication with Humans Evolved to Emerge Early in Domestic Dogs. *Curr. Biol.* **31**, 3137-3144.e11 (2021).
 45. Kaminski, J. Domestic dogs: Born human whisperers. *Curr. Biol.* **31**, R891–R893 (2021).
 46. Somppi, S., Törnqvist, H., Hänninen, L., Krause, C. & Vainio, O. Dogs do look at images: eye tracking in canine cognition research. *Anim. Cogn.* **15**, 163–174 (2012).
 47. Siniscalchi, M., D'Ingeo, S., Minunno, M. & Quaranta, A. Communication in Dogs. *Animals* **8**, 131 (2018).

REVIEWERS' COMMENTS:

Reviewer #2 (Remarks to the Author):

In the revised version of "Functionally analogous body- and animacy-responsive areas in the dog (*Canis familiaris*) and human occipito-temporal lobe", my previous concerns have been taken into account adequately. I have no further concerns.

This manuscript represents the results of a careful and laborous work. I am sure it will be an intriguing addition to the field and provoke further research in this direction.

Reviewer #2 (Remarks to the Author):

In the revised version of "Functionally analogous body- and animacy-responsive areas in the dog (*Canis familiaris*) and human occipito-temporal lobe", my previous concerns have been taken into account adequately. I have no further concerns.

This manuscript represents the results of a careful and laborious work. I am sure it will be an intriguing addition to the field and provoke further research in this direction.

Author response:

We thank the reviewer for their highly positive assessment of our work and valuable suggestions, which significantly strengthened our paper. The reviewer's confidence in the manuscript's potential to contribute to the field and stimulate further research in this area is highly encouraging.